# DECISION-FOCUSED LEARNING: LEARNING TO RANK BASED ON SAMPLE AVERAGE APPROXIMATION

## ABSTRACT

Decision-Focused Learning (DFL) improves prediction models by directly optimizing the decision quality of downstream optimization problems, where Learning to Rank (LTR) approaches treat the solution set as a ranking set and design surrogate losses based on the objective function.The DFL-LTR method exhibits strong applicability; however, it lacks a specially designed subset construction method, which limits its performance.To address this issue, tailored to the intrinsic characteristics of the DFL-LTR method, we first articulate two fundamental bottlenecks that any ranking subset construction must resolve: (*i*) the infeasibility of fully enumerating the solution space; (*ii*) the resulting upper bound on the loss function family that remains unbroken. To eliminate these limitations, we introduce a ranking subset construction paradigm driven by Sample Average Approximation (SAA). By performing stochastic optimization over minibatches, the proposed method yields an equivalent approximation of the complete solution set, suppresses loss variance to stabilise gradients, and consequently elevates the performance upper bound of the entire method. Our LTR-SAA subset construction module is fully plug-and-play: it introduces no extra hyperparameters, incurs zero additional time complexity, and remains compatible with the entire family of LTR loss functions.In the latest open-source benchmark (comprising 7 optimization problems), our proposed method achieves SOTA decision quality on 5 of these problems. Compared with other DFL and 2-stage methods, it demonstrates significant performance advantages and generality. Code is available at
`https://anonymous.4open.science/r/SAA-LTR-33B0`.

## 1 INTRODUCTION

In real-world scenarios, machine learning (ML) is widely used for decision-making (Isinkaye et al., 2015; Kar et al., 2017) in optimization problems (OP). The goal of is to minimize or maximize the objective function, thereby obtaining optimal decisions. However, parameters in the objective function (e.g., costs, prices) are unknown. These parameters must be estimated using ML models and historical data. Only then can downstream decision-making proceed.

In the predict-then-optimize paradigm (Donti et al., 2017; Elmachtoub & Grigas, 2022), the prediction and the decision-making are two independent stages. First stage: Train an ML model on historical data to predict stochastic parameters of the OP objective function. Second stage: Feed these predictions into a solver to get decisions. However, this method treats parameter errors as independent, ignoring interactions between errors and their impact on OP (Cameron et al., 2022).

To address error accumulation and impacts in the 2-stage paradigm, recent decision-focused learning (also called predict-and-optimize or end-to-end paradigm) research has made significant progress (Agrawal et al., 2019; Pogančić et al., 2019; Zharnagambetov et al., 2023). By customizing prediction models for downstream tasks, better task-specific performance is achievable. Such methods typically require end-to-end differential processing of the entire prediction and optimization process, forming a series of DFL approaches. The quality of decisions generated by the prediction model is directly optimized end-to-end and evaluated via a decision loss function. In a recent benchmark test (Geng et al., 2024), DFL has demonstrated better decision quality than the 2-stage method.

At present, DFL's main challenge is that the derivative of the optimal solution with respect to OP parameters is often undefined or zero, making meaningful gradient estimation difficult and requiring

Figure 1: The DFL: LTR-SAA architecture has two components: end-to-end training with ranking loss and a minibatch SAA-guided ranking subset.

gradient estimation methods in most cases. However, the efficacy of such gradient estimation methods is highly contingent upon the type of optimization problem (Pogančić et al., 2019). The key to addressing this limitation is to enable the DFL method to adapt to any type of downstream optimization problem without manual design of gradients and loss functions. Learning to Rank (LTR) is precisely such a DFL method, whose key idea is as follows: feasible solutions are treated as a ranking set, and feasible solutions together with model parameters are mapped to an objective function, which serves as a scoring function for ranking. At this point, gradients are generated jointly by the objective function mapping and ranking loss, making the method fully applicable to optimization problems of any mathematical form. The learning to rank task was first proposed by Burges et al. (2005), while the first application of learning to rank to DFL methods was conducted by Mulumba et al. (2021) and Mandi et al. (2022), who developed a complete set of surrogate ranking tasks and family of surrogate ranking loss functions.

Since OP problems generally cannot fully enumerate the feasible solution set to form a ranking set, Both Mulumba et al. (2021) and Mandi et al. (2022) adopt a ranking subset construction approach that computes a greedy solution to the prediction problem within each minibatch. However, this subset construction method has significant drawbacks, as the greedy sampling strategy was not specifically designed for the DFL-LTR method but was originally developed for perturbation-based DFL approaches. Consequently, there is a lack of rigorous methodological support in theory for constructing subsets that account for the unique characteristics of the DFL-LTR method, and in practice, the mismatch in subset construction severely limits the method's performance. To address these issues, we propose Decision-Focused Learning: Learning to Rank based on Sample Average Approximation. Our work focuses on developing a ranking subset construction method specifically designed for the DFL-LTR approach, tailored to its unique characteristics. Methodologically, we first clarify the properties of an optimal ranking subset, which serves as the objective for both subset construction and optimization, and we establish fundamental lemmas based on this objective. Subsequently, building upon these objectives and lemmas, we propose an Sample Average Approximation (SAA)-based ranking subset construction method and provide a rigorous theoretical analysis of its performance and efficiency. Experimentally, in the latest benchmark test (Geng et al., 2024), our method achieves the SOTA decision quality on 5 out of the 7 optimization problems. The architecture of our method is illustrated in Figure 1.

The main contributions of this paper are as follows:

(i) This work addresses the theoretical gap in the lack of a dedicated sampling subset designed specifically for the DFL-LTR method. We first clarify the objectives of subset construction and optimization under this method. Guided by these objectives, we provide fundamental lemmas that inform the subset construction process.

(ii) We propose a plug-and-play SAA-based subset construction method whose resulting subsets are strictly aligned with the properties of an optimal ranking subset.

(iii) Theoretically, we prove that our method can enhance the performance upper bound of DFL-LTR without incurring additional time complexity compared to the original method. This is also verified by the excellent performance in benchmark experiments.

## 2 PROBLEM STATEMENT AND BACKGROUND

We first formulate the problem that DFL seeks to address. We consider an OP, denoted as Eq (1).

$$\min f(\mathbf{v}; \mathbf{c}) = \langle \mathbf{c}, g(\mathbf{v}) \rangle, \mathbf{v} \in \mathbb{V}, \tag{1}$$

$$\mathbf{v}^*(\mathbf{c}) = \arg\min_{\mathbf{v} \in \mathbb{V}} f(\mathbf{v}; \mathbf{c}), \tag{2}$$

Here, $\mathbb{V}$ denotes the domain of the decision variable $\mathbf{v}$, $\mathbf{v} \in \mathbb{V}^{\dim(\mathbf{v})} \subseteq \mathbb{R}^{\dim(\mathbf{v})}$, determined by constraints. $\mathbb{C}$ denotes the domain of the coefficient vector $\mathbf{c}$, $\mathbf{c} \in \mathbb{C}^{dim(\mathbf{c})} \subseteq \mathbb{R}^{\dim(\mathbf{c})}$. The objective function is a real-valued function $f \in \mathbb{R}$, mapped from the spaces $\mathbb{V}$ and $\mathbb{C}$: $\mathbb{V} \times \mathbb{C} \to f$. $\mathbf{v}^*(\mathbf{c})$ denotes the optimal solution when the coefficient is $\mathbf{c}$, as Eq (2).

**Linear-in-c formulation and generality.** To facilitate the exposition of subsequent methods, we express the objective function in a linear form with respect to $\mathbf{c}$, as $\langle \mathbf{c}, g(\mathbf{v}) \rangle$. where $g : \mathbb{R}^{dim(\mathbf{v})} \to \mathbb{R}^{dim(\mathbf{c})}$ is an *arbitrary vector-valued mapping* of the decision variable $\mathbf{v}$. It should be noted, however, that it can still represent *any type* of explicit optimization problem, since the type of the OP hinges solely on the form of the decision variable $g(\mathbf{v})$; for instance:

- **Linear programming:** $g(\mathbf{v}) = \mathbf{v}$ (knapsack, shortest path, bipartite matching);
- **Quadratic programming:** $g(\mathbf{v}) = [\mathbf{v}; \mathbf{v} \circ \mathbf{v}]$ (portfolio with variance term);
- **Submodular / set functions:** $g(\mathbf{v})$ returns the indicator vector of the chosen subset.

In the conventional predict-then-optimize paradigm, prediction and decision-making are treated as two disjoint stages. During the prediction stage, an input feature vector $\mathbf{x}$ is used to estimate the unknown coefficient vector $\mathbf{c}$; the resulting estimate $\hat{\mathbf{c}}$ then parameterizes the downstream optimization problem. The value of $\mathbf{c}$ is unknown, but the relationship between $\mathbf{x}$ and $\mathbf{c}$ can be learned from a historical dataset $\mathcal{D} = \{(\mathbf{x}_i, \mathbf{c}_i)\}_{i=1}^N$. We train the predictive model $\mathcal{M}_\theta$ via supervised learning to yield coefficient estimates $\hat{\mathbf{c}} = \mathcal{M}_\theta(\mathbf{x})$, $\theta$ is the parameter of the $\mathcal{M}_\theta$. A common regression loss function $\mathcal{L}_{pred}$ is the mean square error (MSE), as Eq (3).

$$\min_{\mathcal{M}_\theta} \mathbb{E}_{(\mathbf{x}_i, \mathbf{c}_i) \in \mathcal{D}}[\mathcal{L}_{pred}(\hat{\mathbf{c}}_i, \mathbf{c}_i)] = \sum_{i=1}^N (\hat{\mathbf{c}}_i - \mathbf{c}_i)^2 \tag{3}$$

In the predict-and-optimize paradigm of DFL, $\hat{\mathbf{c}}$ is merely an intermediate variable. What needs to be evaluated is the quality of the final output decision $\mathbf{v}(\hat{\mathbf{c}})$. Regret is widely used to evaluate decision quality. It can be derived from the difference in decision quality between the solutions under the estimated coefficients $\mathbf{v}^*(\hat{\mathbf{c}})$ and the true coefficients $\mathbf{v}^*(\mathbf{c})$, as shown in Eq (4).

$$regret(\hat{\mathbf{c}}, \mathbf{c}) = \|f(\mathbf{v}^*(\hat{\mathbf{c}}), \mathbf{c}) - f(\mathbf{v}^*(\mathbf{c}), \mathbf{c})\|. \tag{4}$$

We aim to directly use regret as the decision loss $\mathcal{L}_{dec}$ to train the $\mathcal{M}_\theta$. For regret to be used in backward gradient descent, its exact derivative needs to be obtained. However, due to the discontinuity of the solution space $\mathbb{V}$, regret is discontinuous, and the argmax function is non-differentiable, making it impossible to directly use regret as $\mathcal{L}_{dec}$. n the field DFL, there exists a rich body of supervised learning research aimed at reducing regret, with detailed references available in Appendix B Among existing approaches, DFL methods based on the Learning to Rank (LTR) method stand out as highly promising DFL techniques, attributed to two key strengths: first, their strong adaptability. they do not rely on the specific type of optimization problem (e.g., linear/nonlinear, continuous/discrete); second, their excellent performance on benchmark (Geng et al., 2024), which demonstrates stable and efficient results in mainstream evaluation tasks.

### 2.1 DFL BASED ON LEARNING TO RANK

For an OP, its solution space $\mathbb{V}$ is finite. For $\mathbf{v} \in \mathbb{V}$, under the coefficient vector $\mathbf{c}$, the objective function satisfies an ascending order relationship: $f(\mathbf{v}^{k_1}; \mathbf{c}) \leq f(\mathbf{v}^{k_2}; \mathbf{c}) \leq f(\mathbf{v}^{k_3}; \mathbf{c})... \leq f(\mathbf{v}^{k_{|\mathbb{K}|}}; \mathbf{c})$, where $K = \{k_1, k_2, k_3, ...k_{|\mathbb{K}|}\}$ denotes the index of the feasible solution vector, where its subscript represents the ranking. The objective function ranking under the predicted coefficient vector $\hat{\mathbf{c}} : f(\mathbf{v}^{k'_1}; \hat{\mathbf{c}}) \leq f(\mathbf{v}^{k'_2}; \hat{\mathbf{c}}) \leq f(\mathbf{v}^{k'_3}; \hat{\mathbf{c}})... \leq f(\mathbf{v}^{k'_{|\mathbb{K}|}}; \hat{\mathbf{c}})$ The key idea of learning to rank in the DFL method is that if the ranks with respect to $\hat{\mathbf{c}}$ are identical to those with respect to $\mathbf{c}$, where $k = k'$, the regret is zero. The surrogate task is to learn the ranking of each $\mathbf{v} \in \mathbb{V}$ with respect to $\mathbf{c}$.

## 2.2 Family of Learning to Rank Surrogate Loss Functions

Loss functions for ranking surrogate tasks have been extensively and maturely studied. Mandi et al. (2022) extended this family to DFL scenarios by formulating the objective as a scoring function and deriving *pointwise*: Eq. (5), *pairwise*: Eq. (6), and *listwise*: Eq. (7) these ranking variants. In Eq. (6), $\mathbb{V}_O = \{(\mathbf{v}^{k_1}, \mathbf{v}^{k_2}), (\mathbf{v}^{k_1}, \mathbf{v}^{k_3})..., (\mathbf{v}^{k_1}, \mathbf{v}^{k_{|\mathbb{K}|}})\}$ represents the set of ranking pairs corresponding to the optimal solution, and $\tau$ in Eq. (7) denotes a temperature parameter. Since the family of loss functions under the DFL-LTR method is not the focus of our attention, only a brief demonstration is provided.

$$\mathcal{L}_{pointwise}(\hat{\mathbf{c}}, \mathbf{c}, \mathbb{V}) = \frac{1}{|\mathbb{V}|} \sum_{\mathbf{v} \in \mathbb{V}} (f(\mathbf{v}; \mathbf{c}) - f(\mathbf{v}; \hat{\mathbf{c}}))^2. \tag{5}$$

$$\mathcal{L}_{pairwise}(\hat{\mathbf{c}}, \mathbf{c}, \mathbb{V}) = \frac{1}{|\mathbb{V}_O|} \sum_{(p,q) \in \mathbb{V}_O} ((f(\mathbf{v}^p; \hat{\mathbf{c}}) - f(\mathbf{v}^q; \hat{\mathbf{c}}))^2 - (f(\mathbf{v}^p; \mathbf{c}) - f(\mathbf{v}^q; \hat{\mathbf{c}})))^2. \tag{6}$$

$$\mathcal{L}_{listwise}(\hat{\mathbf{c}}, \mathbf{c}, \mathbb{V}) = \frac{1}{|\mathbb{V}|} \sum_{\mathbf{v} \in \mathbb{V}} p_\tau(\mathbf{v} \mid \mathbf{c}) \log p_\tau(\mathbf{v} \mid \hat{\mathbf{c}}), p_\tau(\mathbf{v} \mid \mathbf{c}) = \frac{\exp\left(-\frac{f(\mathbf{v}; \mathbf{c})}{\tau}\right)}{\sum_{\mathbf{v}' \in \mathbb{V}} \exp\left(-\frac{f(\mathbf{v}', \mathbf{c})}{\tau}\right)}. \tag{7}$$

## 3 Methodology

### 3.1 Motivation

To distinguish it from traditional LTR, within the context of DFL, the sets being ranked consist of feasible solutions. Enumeration of all solutions is infeasible due to the continuous nature of solution spaces and the exponential growth of discrete solution spaces with problem scales. Consequently, selecting a high-quality ranking subset $\mathbb{S}$ to enhance method performance emerges as a critical challenge. We summarize the properties of an excellent $\mathbb{S}$, which also serve as its optimization objectives, as follows:

(*i*) It should approximate the full solution set as closely as possible, thereby achieving performance comparable to training with the complete solution set.

(*ii*) Since subset construction methods must seamlessly integrate with existing loss function families and potentially new ones developed in the future, achieving superior performance across all loss functions remains highly challenging. Conversely, in practical applications where multiple loss functions are available, the one yielding optimal performance is typically selected. Thus, the ability of a ranking subset to elevate performance upper bounds holds greater significance.

**Current limitations and theoretical gaps in ranking subset construction within the DFL-LTR method.** Existing DFL-LTR loss function families, proposed by Mandi et al. (2022), directly adopt the solution subset sampling strategy under and introduced by Mulumba et al. (2021). However, Mulumba et al. (2021)s' subset construction method was originally designed for perturbation-based DFL approaches, not specifically for LTR. This mismatch results in a lack of theoretical foundations for ranking subset construction within the DFL-LTR method, hindering the full realization of its performance potential.

**Remark.** Addressing the identified issues in ranking subset construction and aligned with optimization objectives, we supplement and refine the theoretical underpinnings, and propose a Sample Average Approximate (SAA) ranking subset $\mathbb{S}$ construction method. This approach effectively elevates the performance upper bounds of LTR methods within the DFL paradigm.

### 3.2 Basic lemma under the subset optimization objective

**Lemma 1. Impact of rank subsets on the lower bound of the regret.** For a rank subset $\mathbb{S} \subseteq \mathbb{V}$, let $\mathbf{v}^{k_1} = \arg\min_{\mathbf{v} \in \mathbb{S}} f(\mathbf{v}; \mathbf{c})$ denote the optimal solution under the true parameter $\mathbf{c}$, and $\mathbf{v}^{k_m} = \arg\min_{\mathbf{v} \in \mathbb{S}} f(\mathbf{v}; \hat{\mathbf{c}})$ denote the optimal solution under the predicted parameter $\hat{\mathbf{c}}$. Then: (*i*) The necessary and sufficient condition for zero regret: $f(\mathbf{v}^{k_1}; \hat{\mathbf{c}}) \leq f(\mathbf{v}; \hat{\mathbf{c}}) \iff regret(\hat{\mathbf{c}}, \mathbf{c}) = 0$; (*ii*) The lower bound of regret: $r_{\text{lb}}(\hat{\mathbf{c}}, \mathbf{c}) = f(\mathbf{v}^{k_m}; \mathbf{c}) - f(\mathbf{v}^{k_1}; \mathbf{c})$.

**Proof:** If $f(\mathbf{v}^{k_1}; \hat{\mathbf{c}}) \leq f(\mathbf{v}; \hat{\mathbf{c}})$ holds for all $\mathbf{v} \in \mathbb{V}$, then the optimal solution under $\hat{\mathbf{c}}$, denoted $\mathbf{v}^*(\hat{\mathbf{c}})$, is equal to $\mathbf{v}^{k_1}$, which is also the optimal solution under $\mathbf{c}$ (i.e., $\mathbf{v}^*(\mathbf{c}) = \mathbf{v}^{k_1}$). By the definition of regret, The Proposition (*i*) also holds. Thus, the condition is necessary and sufficient. Under the true parameter $\mathbf{c}$, there must exist an optimal solution $\mathbf{v}^{k_1}$ within the $\mathbb{S}$; under the predicted parameter $\hat{\mathbf{c}}$, the optimal solution within $\mathbb{S}$ is $\mathbf{v}^{k_m}$. Since the ranking result is deterministic, the regret can be directly computed as: $regret(\hat{\mathbf{c}}, \mathbf{c}) = f(\mathbf{v}^{k_m}; \mathbf{c}) - f(\mathbf{v}^{k_1}; \mathbf{c})$ From Proposition (*i*) above, regret is zero if and only if $\mathbf{v}^{k_m} = \mathbf{v}^{k_1}$.

**Lemma 2. Optimal equivalent substitute subset.** Given the predicted parameter $\hat{\mathbf{c}}$, suppose the full solution set $\mathbb{V}$ is sorted in ascending order by $f(\cdot; \hat{\mathbf{c}})$ as $\mathbf{v}^{k'_1} \leq \cdots \leq \mathbf{v}^{k_1} \leq \cdots \leq \mathbf{v}^{k'_{|\mathbb{K}|}}$. Define the substitute subset: $\mathbb{S}^* = \{\mathbf{v}^{k_1}\} \cup \{\mathbf{v} \in \mathbb{V} \mid f(\mathbf{v}; \hat{\mathbf{c}}) \leq f(\mathbf{v}^{k_1}; \hat{\mathbf{c}})\}$ If the regret over $\mathbb{S}^*$ is zero, then the regret over the full solution set $\mathbb{V}$ must also be zero.

**Proof:** The subset $\mathbb{S}^*$ consists of all elements in $\mathbb{V}$ whose scores (under $f(\cdot; \hat{\mathbf{c}})$) are no higher than that of $\mathbf{v}^{k_1}$. Notably, $\mathbb{S}^*$ and $\mathbb{V}$ share the same minimum value under $f(\cdot; \hat{\mathbf{c}})$, and this minimum value is achieved by $\mathbf{v}^{k_1}$ in both sets. Consequently, the condition for zero regret over $\mathbb{S}^*$ is identical to that over $\mathbb{V}$ (i.e., $\mathbf{v}^{k_1} = \mathbf{v}^{k'_1}$). Hence, the proposition holds.

**Remark.** Motivated by the optimization objective over $\mathbb{S}$, we first present two intuitively guided lemmas. These lemmas establish fundamental theoretical guarantees for the subsequent design of $\mathbb{S}$ construction methods.

## 3.3 MINIBATCH SAA-GUIDED RANKING SUBSET

---

**Algorithm 1** Backward gradient descent algorithm under the our LTR - SAA method

---

**Require:** $\mathcal{D} = \{(\mathbf{x}_i, \mathbf{c}_i)\}_{i=1}^N$
**Ensure:** Trained model $\mathcal{M}_\theta$
1: Initialize $\mathcal{M}_\theta$, $\mathcal{L} \in \{\text{Eq }(5, 6, 7)\}$, $\mathbf{c}_{\text{rec}} \leftarrow 0$, $\mathbb{S} \leftarrow \{\mathbf{v}_i^*(\mathbf{c}_i) \mid (\mathbf{x}_i, \mathbf{c}_i) \in \mathcal{D}\}$, $L \leftarrow 0$
2: **for** each epoch **do**
3:     **for** each $(\mathbf{x}_i, \mathbf{c}_i)$ **do**
4:         $\mathbf{c}_{\text{rec}} \leftarrow \mathbf{c}_{\text{rec}} + \mathbf{c}_i$
5:         $\sum_{t'=1}^t \sum_i^\beta \mathbf{c}_{i=1} \leftarrow \mathbf{c}_{\text{rec}}$
6:         Solve Eq (9) for $\mathbf{v}_{t,\beta}^{\text{SAA}}$
7:         $\mathbb{S} \leftarrow \mathbb{S} \cup \{\mathbf{v}_{t,\beta}^{\text{SAA}}\}$
8:         $L \leftarrow L + \mathcal{L}(\hat{\mathbf{c}}_i = \mathcal{M}_\theta(\mathbf{x}_i), \mathbf{c}_i, \mathbb{S})$
9:     **end for**
10:     $\theta \leftarrow \theta - \lambda \frac{\partial L}{\partial \theta}$ #Backpropagation
11: **end for**
12: **return** $\mathcal{M}_\theta$

---

**Initialization method for rank subsets.** By Lemma 1, to make the learning lower bound of the optimization objective $regret$ be zero, the true optimal solution $\mathbf{v}(\mathbf{c})$ must lie in the ranking subset every time the loss is calculated. For convenience, we initialize rank subset $\mathbb{S}$ directly based on historical data at the beginning of training, as $\mathcal{D} = \{(\mathbf{x}_i, \mathbf{c}_i)\}_{i=1}^N$.

After the initialization of the rank subset, how to continuously optimize the subset around the optimization objective during training becomes a key problem to be solved subsequently. It should be noted that the methods and experiments in Mandi et al. (2022) and Geng et al. (2024) both adopt the setting of *batch size* = 1. To ensure the fairness of experimental comparisons without reducing training efficiency, this paper defaults to following this batch size. which will be uniformly referred to as minibatch in the subsequent content. Correspondingly, we refer to the $i$-th sample trained in each epoch as the $\beta$-th minibatch.

**Superiority of rank subset construction via Sample Average Approximation method.** First, we review the existing construction method for the rank subset during training in Mandi et al. (2022), along with its limitations. Specifically, for each minibatch, this method adopts a single-point greedy strategy to select a solution $\mathbf{v}^*(\hat{\mathbf{c}})$ (where $\hat{\mathbf{c}} = \mathcal{M}_\theta(\mathbf{x})$, $\hat{\mathbf{c}} = \mathbf{c} + \epsilon$) and adds it to rank subset $\mathbb{S}$. The advantage of this sampling method is that, for each minibatch, it can at least ensure that besides the true optimal solution $\mathbf{v}^*(\mathbf{c})$, there exists another solution $\mathbf{v}^*(\hat{\mathbf{c}})$ that lies within the optimal equivalent substitute subset $\mathbb{S}^*$ defined in Lemma 2). However, the drawbacks of this method are equally prominent: it only focuses on the local optimal solution within a single minibatch, without considering the entire training process. It results in the rank subset $\mathbb{S}$ failing to be optimized toward the optimal subset objective throughout training. Specifically, for each minibatch, the greedy solution $\mathbf{v}(\hat{\mathbf{c}})$ is directly affected by a single predicted parameter $\hat{\mathbf{c}}$ (which may contain outliers or large fluctuations), leading to the following issues: (*i*) The ranking loss exhibits large oscillations, with a significant increase in loss variance; this destabilizes gradient updates and ultimately impacts the theoretical upper bound of training performance; (*ii*) fails to optimize toward the objective of including as many solutions as possible into the optimal equivalent substitute subset $\mathbb{S}^*$, thereby

causing invalid ranking relationships within the rank subset $\mathbb{S}$ (i.e., the relative order of some elements does not satisfy the constraints of $\mathbb{S}^*$ specified in Lemma 2). To address the aforementioned issues and guided by the $\mathbb{S}$ optimization objective, we propose an $\mathbb{S}$ construction method based on Sample Average Approximation (SAA), as shown in Eq (8).

$$\min_{\mathbf{v} \in \mathbb{V}} \mathbb{E}_{\epsilon}[f(\mathbf{v}, \mathbf{c} + \epsilon)] = \sum_{t'=1}^{t} \sum_{i=1}^{\beta} \langle \mathbf{c}_i, g(\mathbf{v}) \rangle = \langle \sum_{t'=1}^{t} \sum_{i=1}^{\beta} \mathbf{c}_i, g(\mathbf{v}) \rangle = f(\mathbf{v}; \sum_{t'=1}^{t} \sum_{i=1}^{\beta} \mathbf{c}_i), \quad (8)$$

$$\mathbf{v}_{t,\beta}^{\text{SAA}} = \arg \min_{\mathbf{v} \in \mathbb{V}} f(\mathbf{v}; \sum_{t'=1}^{t} \sum_{i=1}^{\beta} \mathbf{c}_i), \quad (9)$$

Here, the training epochs are set as $t \in \{0, 1, 2, \ldots, T\}$, where $T$ denotes the preset total number of epochs. The total number of samples in each training epoch is $|\mathcal{D}|$, We add $\mathbf{v}_{t,\beta}^{\text{SAA}}$ from Eq. (9) to set $\mathbb{S}$ in each minibatch. SAA (Kleywegt et al., 2002) is a concise and efficient stochastic programming method. Its key lies in treating the construction and optimization of $\mathbb{S}$ as a stochastic optimization throughout the entire training process, with the following specific and significant advantages:

- (*i*) **Approximation of the $\mathbb{S}^*$.** As Lemma 2 shows, we expect that for each calculation of $\mathcal{L}(\hat{\mathbf{c}}, \mathbf{c}, \mathbb{S})$, the solutions in $\mathbb{S}$ are optimal under $\hat{\mathbf{c}}$, only then can $\mathbb{S}$ approximate $\mathbb{S}^*$. For the decision to be optimal under the perturbed parameter $\hat{\mathbf{c}} = \mathbf{c} + \epsilon$, the most intuitive approach is stochastic optimization, as Eq (8).

- (*ii*) **One-line code modification and plug-and-play support.** We only replace the greedy solution in the original method with the SAA solution. The design of our proposed method introduces no additional time complexity or hyperparameters, and is compatible with existing and future newly developed LTR loss function families; in practical applications, only extra space $\mathbf{c}_{\text{rec}}$ is required to record the set $\sum_{t'=1}^{t} \sum_{i=1}^{\beta} \mathbf{c}_i$, and such constant space complexity is fully acceptable. The specific algorithm flow is shown in Algorithm 1.

- (*iii*) **Reduced variance of ranking loss and improved upper bound of performance.** The ranking score variance decays as $\mathcal{O}(1/(t|\mathcal{D}| + \beta))$ along training, yielding higher-probability correct pairs. The cumulative cost $\bar{\mathbf{c}}_{t,\beta}$ is an *unbiased* surrogate of the population cost $\mathbf{c}$; its solution is on average $\mathcal{O}(1/\sqrt{t})$-closer to the true optimal face of polyhedron $\mathbb{V}$, tightening the regret upper bound.

Advantages (*i*) and (*ii*) here can be directly derived from the stochastic programming approach under SAA and the definition of time complexity, respectively, and are explained in detail in Appendix D. Here, we provide an explanation of the complexity result for Advantage (*iii*).

**Assumption 1: Sub-Gaussian cost noise.** For every sample $(\mathbf{x}, \mathbf{c})$ the residual $\varepsilon = \mathbf{c} - \mathbb{E}[\mathbf{c} \mid \mathbf{x}]$ is coordinate-wise $\sigma$-sub-Gaussian: $\mathbb{E}\left[e^{\lambda \varepsilon_j}\right] \leq e^{\lambda^2 \sigma^2 / 2}, \quad \forall \lambda \in \mathbb{R}, \forall j = 1, \ldots, p$.

It is important to note that based on the assumption in Assumption 1 that historical labels $\mathbf{c}_i$ are independently and identically distributed (i.i.d.), the following conclusion can be derived: although the learnable parameter $\theta_t$ evolves non-stationarily, the recommended cost estimate $\mathbf{c}_{\text{rec}}$ remains an unbiased estimate of the population cost at each epoch $t$.

**Lemma 3: Variance reduction via cumulative minibatch.** Algorithm 1 maintains the cumulative cost $\bar{\mathbf{c}}_{t,\beta} = \sum_{t'=1}^{t} \sum_{i=1}^{\beta} \mathbf{c}_i / (t|\mathcal{D}| + \beta)$. For any fixed $\mathbf{v} \in \mathbb{V}$ and any training epoch $t \geq 0$ and $\beta \geq 1$, $\text{Var}[\langle \bar{\mathbf{c}}_{t,\beta}, g(\mathbf{v}) \rangle] = g(\mathbf{v})^{\top} \Sigma g(\mathbf{v}) / (t|\mathcal{D}| + \beta) \leq \sigma^2 \|g(\mathbf{v})\|^2 / (t|\mathcal{D}| + \beta)$. Hence, the rank loss variance decays as $\mathcal{O}(1/(t|\mathcal{D}| + \beta))$ along training.

**Proof:** Since $\bar{\mathbf{c}}_{t,\beta}$ is an independent and identically distributed average of $t$ zero-mean vectors $\varepsilon_i$, $\text{Cov}[\bar{\mathbf{c}}_{t,\beta}] = (t|\mathcal{D}| + \beta)\Sigma / (t|\mathcal{D}| + \beta)^2 = \Sigma / (t|\mathcal{D}| + \beta)$. Substitute into the quadratic form and follow from Assumption 1, we have: $\text{Var}\left[\langle \bar{\mathbf{c}}_{t,\beta}, g(\mathbf{v}) \rangle\right] = g(\mathbf{v})^{\top} \text{Cov}[\bar{\mathbf{c}}_{t,\beta}] g(\mathbf{v}) = g(\mathbf{v})^{\top} \Sigma g(\mathbf{v}) / (t|\mathcal{D}| + \beta) \leq \sigma^2 \|g(\mathbf{v})\|^2 / (t|\mathcal{D}| + \beta)$.

**Lemma 4: Solution error bound.** Let $W = \max_{\mathbf{v}, \mathbf{v}' \in \mathbb{V}} \|g(\mathbf{v}) - g(\mathbf{v}')\|$ be the $g$-diameter of $\mathbb{V}$. With probability at least $1 - \delta$ over the randomness of the first $(t|\mathcal{D}| + \beta)$ samples, $\|g(\mathbf{v}_{t,\beta}^{\text{SAA}}) - g(\mathbf{v}^{\text{true}})\| \leq 2\sigma W \sqrt{(2 \log(2/\delta))/(t|\mathcal{D}| + \beta)}$. Thus the SAA solution approaches the true optimal face at $\mathcal{O}(1/\sqrt{t|\mathcal{D}| + \beta})$ rate.

**Proof:** By Assumption 1 and union bound, $\|\bar{\mathbf{c}}_{t,\beta} - \mathbf{c}\|_\infty \leq \sigma\sqrt{2\log(2p/\delta)/(t|\mathcal{D}| + \beta)}$ with probability $\geq 1 - \delta$. Since $f(\mathbf{v}, \cdot)$ is linear, uniform concentration over the finite polyhedron $\mathbb{V}$ gives $\max_{\mathbf{v} \in \mathbb{V}} \|\langle\bar{\mathbf{c}}_{t,\beta}, g(\mathbf{v})\rangle - \langle\mathbf{c}, g(\mathbf{v})\rangle\| \leq \sigma W\sqrt{(2\log(2p/\delta))/(t|\mathcal{D}| + \beta)}$. The result follows from standard perturbation bounds for linear programs.

**Theorem 1: PAC regret of LTR-SAA.** Run Algorithm 1 for $T$ epochs with historical data size $|\mathcal{D}|$ and learning rate $\lambda = \Theta(1/\sqrt{T})$. Under Assumption 1 and $\|g(\mathbf{v})\| \leq G$, with probability $\geq 1 - \delta$:

$$regret(T) \leq \underbrace{\frac{2\sigma GW}{\sqrt{|\mathcal{D}|}}\sqrt{\frac{2\log(2/\delta)}{T}}}_{\text{subset error}} + \underbrace{\mathcal{O}\left(\frac{G^2\log|\mathbb{V}|}{\sqrt{T}}\right)}_{\text{learning error}}.$$

Hence increasing the number of training epochs $T$ provably tightens the regret upper bound at an $O(1/\sqrt{T})$ rate.

**Proof:** Decompose regret into subset error due to using Lemma 4 and classical online learning regret of ranking over steps. Standard online convex optimisation yields the second term.

## 4 EXPERIMENTS

### 4.1 BENCHMARK EXPERIMENTAL SETUP

To verify the effectiveness of the proposed method, we conducted a series of experiments on the latest open-source benchmark framework (Geng et al., 2024).

**Datasets and OP.** This benchmark experiment aims to test the performance of the DFL method on different types of optimization problems, including integer linear programming (ILP), quadratic programming (QP), submodular, and top-k problems, with a total of 7 optimization problems. These problems are based on operational optimization scenarios and data in real-world production. The data source (DS) include: **SEMO**: Irinn et al. (2012), **Generated**: Elmachtoub & Grigas (2022), **Quanal**: Quandl (2022), **Cora**: Sen et al. (2008), and **Yahoo**: Yahoo (2007). These problems are as follows: (1) Knapsack (Gen), type: ILP, DS: SEMO; (2)Knapsack (Energy), type: ILP, DS: Generated; (3) Scheduling (Energy), type: ILP, DS: Generated; (4) Budget allocation, type: Submodular, DS: Yahoo; (5) TopK, type: Top-K, DS: Generated; (6) Bipartite Matching, type: ILP, DS: Cora; (7) Portfolio Optimization, type: QP, DS: Quanal. The real-world background, specific mathematical forms, datase licenses for these OP are detailed in Appendix E.

**Comparative Methods.** In the benchmark on the above 7 optimization problems, a total of 10 DFL methods and the traditional 2-stage method (serving as the baseline) were tested. These 10 DFL methods include the original **DFL-LTR**(Mandi et al., 2022): **Org-Pt, Org-Pr** and **Org-Lt**. The remaining methods are as follows: **NCE**: Mandi et al. (2022), **LODL**: Shah et al. (2022), **SPO**: Elmachtoub & Grigas (2022), **CPLayer**: Agrawal et al. (2019), **DFL**: Shah et al. (2022), **Identity**: Sahoo et al. (2023) and **Blackbox**: Pogančić et al. (2019). All these DFL methods have been published in top ML conferences or journal, with detailed descriptions provided in Appendix E.

**Benchmarking Protocols.** To guarantee a rigorously fair comparison, we adopt the *exact protocol of the original benchmark* Geng et al. (2024): every parameter, model, and hyper-parameter that could influence the outcome is held fixed, and only the DFL method is swapped for our SAA-LTR variants (SAA-Pt, SAA-Pr, SAA-Lt). *Seed fixed:* all random seeds are fixed to 2023; hence we report *single-run metrics without standard deviation*. The sole deviation is the use of updated hardware, which merely shortens training time and leaves the results unchanged. When comparing training times, all measurements are reported on our hardware; The full configuration of the benchmark is provided in Appendix C.

### 4.2 EXPERIMENT RESULTS AND ANALYSIS

**Decision Quality and Applicability .** Since the LTR framework encompasses multiple loss functions, we adopt an ensemble optimization strategy: we screen models trained with different loss functions on the validation set, select the one with the lowest regret, and its validation set performance is presented in Table 1. Based on this optimal ensemble model, we conduct further tests

| Methods | Knapsack (Gen) | Knapsack (Energy) | Scheduling (Energy) | Budget Allocation | TopK (Cubic) | Bipartite Matching | Portfolio |
|---|---|---|---|---|---|---|---|
| Org-Pt | 2.894 | 78.123 | 19923.463 | 0.073 | 0.078 | 35.500 | 0.203 |
| Org-Pr | 2.933 | 83.247 | 15823.189 | 0.007 | 0.286 | 35.032 | 0.233 |
| Org-Lt | 2.662 | 77.345 | 15572.437 | 0.008 | 0.032 | 34.893 | 0.289 |
| SAA-Pt (ours) | **2.438** | 77.256 | 129955.617 | 0.064 | **0.009** | 34.500 | 0.334 |
| SAA-Pr (ours) | 2.773 | 91.002 | 38020.166 | 0.007 | 0.175 | **33.250** | 0.302 |
| SAA-Lt (ours) | 2.873 | **76.695** | **22572.967** | **0.006** | 0.044 | 33.500 | **0.240** |

Table 1: The regret of models corresponding to different loss functions under the LTR framework on the validation set of each optimization problem. Among them, the indices underlined represent the optimal model performance under the traditional Org-LTR framework, and the indices **bolded** denote the optimal model performance under the proposed SAA-LTR framework. We select the optimal models from each framework as their respective representatives for subsequent tests.

| Methods | Knapsack (Gen) | Knapsack (Energy) | Scheduling (Energy) | Budget Allocation | TopK (Cubic) | Bipartite Matching | Portfolio |
|---|---|---|---|---|---|---|---|
| 2-stage | 6.595 | 8.745 | 1.793 | 20.332 | 0.110 | 92.963 | 0.243 |
| DFL | 11.744 | 8.353 | 6.272 | 35.970 | 1.974 | 91.364 | 0.380 |
| Blackbox | 24.274 | 35.705 | 6.503 | 26.905 | 13.944 | 91.988 | 0.286 |
| Identity | 31.874 | 17.156 | 5.690 | 14.799 | 13.944 | 91.868 | 0.280 |
| CPLayer | 24.769 | 36.402 | – | – | – | 92.007 | 0.309 |
| SPO | 6.223 | 8.407 | **1.505** | 5.559 | 160.408 | 93.327 | 0.245 |
| LODL | 6.044 | 9.567 | 1.786 | 25.700 | 0.172 | 91.113 | **0.160** |
| NCE | 13.438 | 11.932 | 1.663 | 9.979 | 160.408 | 92.662 | 0.367 |
| Org-LTR | 6.031 | 8.083 | 1.540 | 5.742 | 0.193 | 91.0.35 | 0.214 |
| SAA-LTR (ours) | **5.907** | **8.007** | 2.339 | **4.259** | **0.082** | **88.211** | 0.333 |

Table 2: Decision quality metrics for each method across problems in benchmark, the relative regret (w.r.t. optimal objective %) on the test set is presented as follows. The **best** method is **bolded and underlined** in each column. "-" means non-applicable. All results except those obtained under our method in the table are identical to the original values reported by Geng et al. (2024) (including those marked "non-applicable"), as the benchmark uses a single, fixed random seed that guarantees reproducibility. The reasons for the "non-applicable" entries are explained in the original benchmark.

on the benchmark test set. For the evaluation metric of decision quality, we adopt relative regret consistent with the benchmark, which is calculated on the test set as: average regret divided by average optimal objective function value. As shown in Table 2, our proposed method achieves SOTA decision quality in 5 out of the 7 optimization problems in the benchmark, delivering an exceptionally impressive performance. In the original benchmark test, even the best DFL methods (e.g., original LTR and SPO) only achieved optimal decision-making in 2-3 problems. Thus, our method not only achieves optimal performance on individual problems but also significantly outperforms other DFL methods in terms of usability and scalability when dealing with different types of optimization problems. It is worth noting that for the TopK (cubic) problem, the SOTA method in the original benchmark was the traditional 2-stage method, which was the only problem where the 2-stage method outperformed DFL methods. However, our SAA-pt method successfully surpassed the 2-stage method in decision quality, and since then, DFL methods in the benchmark have achieved an overall lead over the 2-stage method. Within the LTR method, for the two problems where SOTA performance remains unachieved, traditional methods also struggle to break through existing bottlenecks. Notably, our approach consistently maintains competitiveness relative to traditional methods with respect to the upper bound of decision quality. This indicates that the DFL-LTR, enhanced by our theoretical advancements, has reached the SOTA level in the current method.

For the subsequent visualization analysis, we select three representative problems of different types and sizes (*Kn (Gen)*: Knapsack (Gen), ILP, small size; *Cub*: TopK (Cubic), TopK, medium size; *BA*: Budget Allocation, Submodular, large size). The complete experimental results are provided in Appendix F, which are consistent with the findings discussed in the main text.

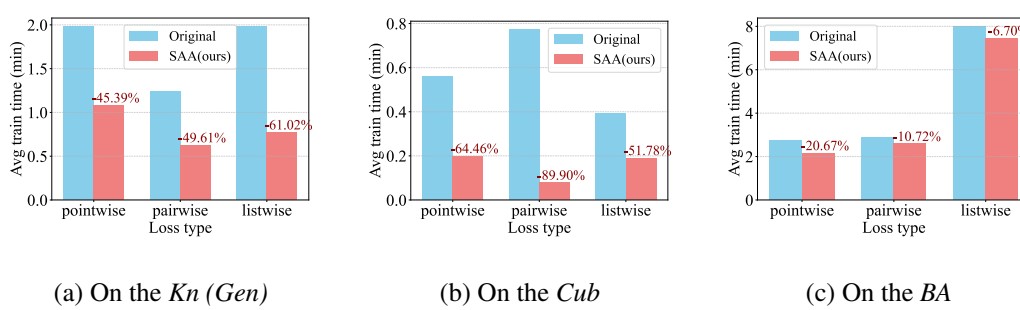

(a) On the *Kn (Gen)*          (b) On the *Cub*          (c) On the *BA*

Figure 2: Training efficiency comparison with the original method on problems of different types and scales, using loss familys. The percentages in the figure represent the relative differences.

**Training Efficiency.** We continue to use the average training time per epoch from the original benchmark as the metric for training efficiency. When designing LTR-SAA, we ensured that performance upper bound improvements would not come at the cost of increased algorithmic time complexity. Theoretically, as shown in Appendix D, we prove that its time complexity does not exceed that of the original greedy subset construction method. Experimentally, we demonstrate training efficiency across three optimization problems and three loss functions (see Figure 2). The results are surprising: rather than matching the original method's average training time as expected, our approach achieves significant time savings. This can be attributed to two factors: first, the SAA method converges to one or more optimal stochastic solutions through sampling, and union-based deduplication reduces the size of our ranking subsets compared to the non-convergent greedy subsets; second, the computational complexity of loss functions is linear with respect to subset size. Consequently, our method exhibits superior training efficiency in experiments.

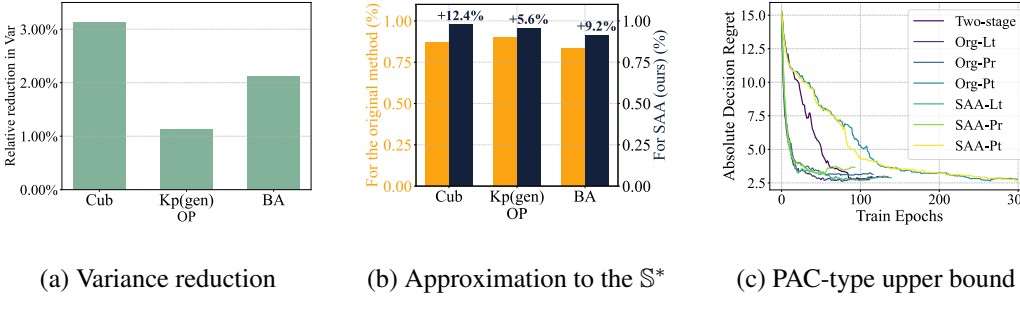

(a) Variance reduction          (b) Approximation to the $\mathbb{S}^*$          (c) PAC-type upper bound

Figure 3: Experimental verification of the theoretical properties of the LTR-SAA

**Verification of ranking score variance reduction and approximation to the $\mathbb{S}^*$.** Since both the variance of ranking scores on each minibatch and the degree of approximation to the $\mathbb{S}^*$ during training affect model updates, we evaluate two metrics across minibatches after training: (*i*) the average relative reduction in ranking score variance compared with the original subset (see Figure 3 (a)); ((*ii*) the proportion of elements from the $\mathbb{S}^*$ that are retained in the selected subset (see Figure 3 (b)). The results indicate that our method does not significantly reduce ranking score variance, contributing minimally to performance improvement, but achieves a modest improvement in approximating the $\mathbb{S}^*$. In summary, while our method achieves the intended goals of variance reduction and the $\mathbb{S}^*$ approximation, its overall impact on model performance remains limited. We find that the primary driver of performance improvement is the improved approximation of $\mathbb{S}^*$ , rather than variance reduction,which was initially regarded as a key advantage. This is essentially because approximating $\mathbb{S}^*$ can directly reduce the "subset error" term in the PAC regret decomposition of LTR-SAA, while the effect of variance reduction is inherently subtle and fails to directly contribute to tightening the regret upper bound.

**Verification and demonstration of the PAC regret for LTR-SAA.** As shown in Figure 3 (c), the regret on the *Kn (Gen)* problem decreases and converges as the number of training epochs increases. It can be observed that our method continuously tightens the upper bound of regret during training,

resulting in a relatively slower convergence rate but also enabling the regret to consistently reach new optimal values. This experimental result validates the PAC-type upper bound of LTR-SAA, shedding light on the theoretical basis behind its excellent performance bound.

The above results and analyses are applicable to the common characteristics of the problems;The remaining secondary and complete data along with their analyses are listed in Appendix F.

## 5 RELATED WORK

All the work in this paper is built upon the DFL-LTR theoretical method proposed by Mandi et al. (2022), and serves as a refinement and improvement of this theory. Their focus was on introducing LTR methods into the DFL domain and developing a family of loss functions to enable supervised learning. Our work, on the other hand, constructs optimization based on ranked subsets, which better aligns with the characteristics of the DFL domain and thereby improves performance. Other related DFL works are listed in the experimental section, with a detailed review provided in Appendix B.

## 6 CONCLUSION

This paper addresses DFL-LTR subset construction challenges, with key conclusions: It fills the theoretical gap of DFL-LTR-specific sampling subsets by clarifying construction/optimization objectives and establishing guiding lemmas. A plug-and-play SAA-based subset method is proposed. Theoretical analysis confirms enhanced DFL-LTR performance upper bound without extra complexity, validated by benchmark experiments. This work provides a theoretical and methodological solution for DFL-LTR subset construction, enriching its system and aiding performance. However, our method still has limitations: ($i$) Our LTR-SAA is an aggressive method that pursues the performance upper bound, and it cannot guarantee improvements in average performance or stability across all loss function families; ($ii$) As an ensemble framework based on selecting the optimal model, LTR still suffers from the efficiency disadvantage of requiring additional training and evaluation of multiple models compared to other DFL methods, which our method fails to address. Nevertheless, it is worth noting that our method also inspires a research direction that combines two data-driven optimization paradigms: DFL and stochastic optimization. This is because DFL can obtain additional raw feature information, while stochastic optimization offers statistical interpretability in decision-making. If the research outcomes of these two currently independent fields can be integrated, it will be an interesting and promising research direction.

## Reproducibility Statement

Our experimental results are fully reproducible. We provide detailed descriptions of the dataset sources, hardware and software configurations of the experimental equipment, fixed random seed settings, and hyperparameter settings in both the main text and the Appendix C and E. An anonymous link to the code repository is provided in the abstract, and an identical code archive is also submitted as supplementary material. Notably, our experiments are conducted on existing open-source benchmarks, and we strictly follow the benchmark protocols. With fixed random seeds, the experimental results are reproducible at a single run, enabling direct comparison with the original results and facilitating performance evaluation of the proposed method.

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

# APPENDIX

## A STATEMENT ON THE USE OF LARGE LANGUAGE MODELS

In this research and manuscript preparation, we adhere to academic integrity and regulate Large Language Models (LLMs) use as follows:

**Authorized Use.** Writing Polishing: LLMs optimized text expression (grammar, terminology, logical flow) to enhance academic rigor/readability. Literature Assistance: LLMs aided literature retrieval (e.g., keyword expansion) and structured abstract summarization, with all results manually verified.

**Prohibited Use LLMs were not used in:** Research conception/design (hypotheses/methods from independent thinking/discussions). Data collection/analysis (all handled manually by researchers). Conclusion deduction/discussion (conclusions/interpretations from our independent judgment).

**Transparency Commitment.** We truthfully disclose LLM use in the paper, ensure assisted content does not compromise originality/authenticity, and uphold academic ethics.

## B RELATED WORK

We first review the relevant studies on DFL methods and categorize them into three types: (*i*) *Discrete methods*: Due to the discreteness of decision variables in OP, the gradient with respect to the coefficients of the objective function is a piecewise constant function, where the gradient is either zero or undefined. Discrete methods first solve the OP in the forward pass using a solver, and in the backward pass, compute the gradient via designed interpolation functions ( Pogančić et al. 2019) or other gradient estimation methods. Representative methods include **DFL**: Shah et al. (2022), **Black-box**: Pogančić et al. (2019), **Perturb**: Berthet et al. (2020), **I-MLE**: Niepert et al. (2021), **Identity**: Sahoo et al. (2023), etc.; (*ii*) *Continuous methods*: These methods focus on computing gradients by estimating the relaxed form of OP. They treat the original problem as a continuous optimization problem, which is essentially implicit differentiation. In the forward pass, the relaxed optimization problem is solved; in the backward pass, gradients are obtained via automatic differentiation (e.g., KKT conditions). Representative methods include OptNet: Amos & Kolter (2017), **CPLayer**: Agrawal et al. (2019), **QPTL**: Wilder et al. (2019), **IntOpt**: Mandi & Guns (2020), **SPO-relax**: Mandi et al. (2020), etc.; (*iii*) *Surrogate methods*: These methods directly bypass gradient estimation and design surrogate loss functions. They typically replace the original optimization objective function with a learned surrogate function. Since the objective function is acquired through learning, it has the significant advantage of inherent differentiability. Representative methods include **LODL**: Shah et al. (2022), **EGL**: Shah et al. (2024), **LANCER**: Zharnagambetov et al. (2023), **SurCO**: Ferber et al. (2023), etc.

The learning to rank method focused on in this paper belongs to the third category of surrogate methods. The reason is that this method acquires the gradient of the original optimization problem by designing a differentiable and learnable surrogate ranking task. The learnable ranking method was first proposed by Burges et al. (2005). They also proposed the pairwise ranking loss. The listwise ranking loss proposed by Cao et al. (2007) further enriches the surrogate ranking loss functions, thus forming a complete family of ranking loss functions. The work most closely related to ours is that of Mulumba et al. (2021) and Mandi et al. (2022), who extended learning to rank to the DFL paradigm. However, there are significant differences between our work and theirs: their focus lies in the generalization of ranking tasks and ranking losses, whereas ours focuses on addressing the problem that the feasible solution set of an OP cannot be fully enumerated. We aim to efficiently search for a ranking subset that is as equivalent as possible to the complete solution set, thereby improving the performance of DFL methods. Our work and theirs are complementary, jointly improving the learning to rank DFL system.

## C   Detailed Configuration of Experimental Setup

### C.1   Experimental equipment and environment

All experiments were conducted on a single Linux workstation. The hardware and core software stack are as follows:

- **Hardware**: Intel Core i5-9400 CPU @ 2.90 GHz, 16 GB RAM, NVIDIA RTX 3080 GPU (12 GB GDDR6X).

- **Operating System**: Ubuntu 16.04 LTS (kernel 4.15.0).

- **Software Stack**: Python 3.9.12, PyTorch 2.4.1+cu118, CUDA 11.8, cuDNN 8.6.0.

- **Solvers & Modeling Layers**: `cvxpylayers` 0.1.6 and `gurobipy` 9.5.2.

The complete `requirements.txt` is provided in the anonymous repository.

### C.2   Details of the Prediction Model and Its Hyperparameters

For the predictive model, we employ a two-layer Multi-Layer Perceptron (MLP) with 32 hidden units in each layer. We adopt the ReLU activation function. Unless stated otherwise, 20% of the training data is extracted for validation purposes. During the training process, the *Adam* Optimizer is adopted, and a grid search is performed over the learning rate values $\{0.1, 0.05, 0.01, 0.005, 0.001\}$. It is noteworthy that this optimization module (encompassing the MLP architecture, validation split strategy, optimizer, and learning rate search scheme) is uniformly utilized across all seven of our optimization problems. The **random seed** is set to **2023**.

### C.3   End-to-End Training Setup and Corresponding Hyperparameters

Unless otherwise specified, 20% of the training dataset is reserved for validation. In each experimental run, models are trained for 300 epochs. Training terminates early if the validation set shows no improvement in regret for 50 consecutive epochs. The epoch yielding the minimum regret (or maximum uplift) on the validation set is selected for testing. For batch size, we use a default of 256, with a special case: due to the unique characteristics of the learning to rank method, its batch size is set to 1.

## D   Methodological Theoretical Supplements

### D.1   Time Complexity Analysis

The time complexity of the modified algorithm remains identical to the original one, as elaborated below. The key of time complexity lies in the **sample processing time** within the nested loops, combined with the scales of epochs and sample size.

#### D.1.1   Original Algorithm Analysis

For each sample $(\mathbf{x}_i, \mathbf{c}_i)$ in the original algorithm:

- The key computational step was solving $\mathrm{E}_{t,\beta}[\cdot]$ (via sample average approximation), which took $\mathcal{O}(P)$ time (where $P$ denotes fixed computational overhead such as sampling count or iteration steps).

- Other operations (arithmetic calculations and set updates) were constant time $\mathcal{O}(1)$.

Thus, the sample processing time was $\mathcal{O}(P)$.

---

**Algorithm 2** Original backward gradient descent algorithm under the greedy subset method

---

**Require:** $\mathcal{D} = \{(\mathbf{x}_i, \mathbf{c}_i)\}_{i=1}^N$
**Ensure:** Trained model $\mathcal{M}_\theta$
1: Initialize $\mathcal{M}_\theta$, $\mathcal{L} \in \{\text{Eq } (5, 6, 7)\}$,$\mathbf{c}_{\text{rec}} \leftarrow 0$, $\mathbb{S} \leftarrow \{\mathbf{v}_i^*(\mathbf{c}_i) \mid (\mathbf{x}_i, \mathbf{c}_i) \in \mathcal{D}\}$, $L \leftarrow 0$
2: **for** each epoch **do**
3:     **for** each $(\mathbf{x}_i, \mathbf{c}_i)$ **do**
4:         $\hat{\mathbf{c}}_i = \mathcal{M}_\theta(\mathbf{x_i})$
5:         Solve greedy soultion $\mathbf{v}^*(\hat{\mathbf{c}})$
6:         $\mathbb{S} \leftarrow \mathbb{S} \cup \{\mathbf{v}^*(\hat{\mathbf{c}})\}$
7:         $L \leftarrow L + \mathcal{L}(\hat{\mathbf{c}}_i = \mathcal{M}_\theta(\mathbf{x}_i), \mathbf{c}_i, \mathbb{S})$
8:     **end for**
9:     $\theta \leftarrow \theta - \lambda \dfrac{\partial L}{\partial \theta}$ *#Backpropagation*
10: **end for**
11: **return** $\mathcal{M}_\theta$

---

---

**Algorithm 3** Our backward gradient descent algorithm under the SAA subset method

---

**Require:** $\mathcal{D} = \{(\mathbf{x}_i, \mathbf{c}_i)\}_{i=1}^N$
**Ensure:** Trained model $\mathcal{M}_\theta$
1: Initialize $\mathcal{M}_\theta$, $\mathcal{L} \in \{\text{Eq } (5, 6, 7)\}$,$\mathbf{c}_{\text{rec}} \leftarrow 0$, $\mathbb{S} \leftarrow \{\mathbf{v}_i^*(\mathbf{c}_i) \mid (\mathbf{x}_i, \mathbf{c}_i) \in \mathcal{D}\}$, $L \leftarrow 0$
2: **for** each epoch **do**
3:     **for** each $(\mathbf{x}_i, \mathbf{c}_i)$ **do**
4:         $\mathbf{c}_{\text{rec}} \leftarrow \mathbf{c}_{\text{rec}} + \mathbf{c}_i$
5:         $\sum_{t'=1}^t \sum_i^\beta \mathbf{c}_{i=1} \leftarrow \mathbf{c}_{\text{rec}}$
6:         Solve Eq (9) for $\mathbf{v}_{t,\beta}^{\text{SAA}}$
7:         $\mathbb{S} \leftarrow \mathbb{S} \cup \{\mathbf{v}_{t,\beta}^{\text{SAA}}\}$
8:         $L \leftarrow L + \mathcal{L}(\hat{\mathbf{c}}_i = \mathcal{M}_\theta(\mathbf{x}_i), \mathbf{c}_i, \mathbb{S})$
9:     **end for**
10:     $\theta \leftarrow \theta - \lambda \dfrac{\partial L}{\partial \theta}$ *#Backpropagation*
11: **end for**
12: **return** $\mathcal{M}_\theta$

---

Original Algorithm Analysis For each sample $(\mathbf{x}_i, \mathbf{c}_i)$ in the original algorithm:

- The key computational step was solving $\mathrm{E}_{t,\beta}[\cdot]$ (via sample average approximation), which took $\mathcal{O}(P)$ time (where $P$ denotes fixed computational overhead such as sampling count or iteration steps).

- Other operations (arithmetic calculations and set updates) were constant time $\mathcal{O}(1)$.

Thus, the sample processing time was $\mathcal{O}(P)$.

Modified Algorithm Analysis In the modified version, for each sample $(\mathbf{x}_i, \mathbf{c}_i)$:

- The critical step is now "solving for the greedy solution $\hat{v}_{\text{greedy}}$". Greedy solutions typically involve localized decision-making with fixed-step operations (e.g., iterating over a finite candidate set or executing $P'$ greedy selections). If $T'$ is of the same order as $P$ (i.e., $P' = \Theta(T)$), this step maintains a time complexity of $\mathcal{O}(P') = \mathcal{O}(P)$.

- Remaining operations (set updates and loss calculations) remain $\mathcal{O}(1)$.

Hence, the sample processing time remains $\mathcal{O}(P)$.

Overall Complexity With $E$ epochs and $N$ samples, the **overall time complexity** of both algorithms is $O(E \cdot N \cdot P)$, confirming their computational equivalence.

## D.2 Brief Introduction to SAA Stochastic Optimization Method

Sample Average Approximation (SAA) is a classical numerical method for stochastic optimization (Kleywegt et al., 2002): it replaces the expectation term $\mathbb{E}_{\xi \sim P}[f(x, \xi)]$ in the original stochastic optimization problem $\min_{x \in X} \mathbb{E}_{\xi \sim P}[f(x, \xi)]$ (where $X \subseteq \mathbb{R}^d$ is the feasible region, $\xi$ is a random variable with distribution $P$, and $f(x, \xi)$ is the stochastic objective function) with the empirical average of $N$ independent and identically distributed (i.i.d.) samples $\xi_1, \ldots, \xi_N$ from $P$, forming the deterministic approximate problem $\min_{x \in X} \frac{1}{N} \sum_{i=1}^{N} f(x, \xi_i)$ whose solution $x_N^*$ serves as the approximate optimal solution to the original problem; notably, as $N$ increases, $x_N^*$ converges almost surely to the true optimal solution $x^*$ (consistency, guaranteed by the Law of Large Numbers), and for large $N$, the deviation $N^{1/2}(x_N^* - x^*)$ follows an asymptotic normal distribution under stronger conditions, making SAA a reliable approach for scenarios where $P$ is inaccessible or the expectation is hard to compute analytically.

## D.3 Supplemental Proofs for Batch Size $B > 1$

Although traditional DFL-LTR defaults to $batchsize = 1$ (sample-by-sample update), we initially adopted this setting to align with the experimental protocol of the baseline method. However, our proposed LTR-SAA method itself has no dependence on the batch size. Below, we illustrate its applicability in the scenario of $batchsize > 1$ from both implementation and theoretical perspectives:

$(i)$ Seamless integration and straightforward implementation: Assuming the batch size is $B$, the update operation is adjusted to: $c_{rec}+ = c_{batch}$ (where $c_{batch} = \sum_{i=1}^{B} c_i$); the rest of the solution process remains identical to that when batch size = 1.

$(ii)$ Inheritability of theoretical advantages:

**Lemma 3: Variance Reduction via Cumulative Minibatch (Batch Size $B$)** When batch size = $B > 1$, the algorithm maintains the cumulative cost as $\bar{c}_{t,\beta} = \sum_{\tau=1}^{t} \sum_{i=1}^{B} c_i / (t|\mathcal{D}| + \beta B)$. For any fixed $\mathbf{v} \in \mathcal{V}$, any training epoch $t \geq 0$, and $\beta \geq 1$:

$$\mathrm{Var}[\langle \bar{\mathbf{c}}_{t,\beta}, g(\mathbf{v}) \rangle] = g(\mathbf{v})^\top \Sigma g(\mathbf{v}) / (t|\mathcal{D}| + \beta B) \leq \sigma^2 \|g(\mathbf{v})\|^2 / (t|\mathcal{D}| + \beta B)$$

Hence, the rank loss variance decays as $\mathcal{O}(1/(t|\mathcal{D}| + \beta B))$, which is faster than the $\mathcal{O}(1/(t|\mathcal{D}| + \beta))$ rate when $B = 1$.

**Proof** Let $\bar{c}_{t,\beta}$ be the independently and identically distributed average of $t$ minibatches (each minibatch contains $B$ zero-mean vectors $e_i$), i.e., $\bar{c}_{t,\beta} = \sum_{\tau=1}^{t} \sum_{i=1}^{B} e_i / (t|\mathcal{D}| + \beta B)$. Substituting into the quadratic form and following from Assumption 1, we have:

$$\mathrm{Var}[\langle \bar{c}_{t,\beta}, g(\mathbf{v}) \rangle] = g(\mathbf{v})^\top \mathrm{Cov}[\bar{c}_{t,\beta}] g(\mathbf{v}) = g(\mathbf{v})^\top \Sigma g(\mathbf{v}) / (t|\mathcal{D}| + \beta B) \leq \sigma^2 \|g(\mathbf{v})\|^2 / (t|\mathcal{D}| + \beta B)$$

**Lemma 4: Solution Error Bound (Batch Size $B$)** Let $W = \max_{\mathbf{v}, \mathbf{v}' \in \mathcal{V}} \|g(\mathbf{v}) - g(\mathbf{v}')\|$ be the $g$-diameter of $\mathcal{V}$. With probability at least $1 - \delta$ over the randomness of the first $(t|\mathcal{D}| + \beta B)$ samples:

$$\|g(\mathbf{v}_{t,\beta}^{\mathrm{SAA}}) - g(\mathbf{v}^{\mathrm{true}})\| \leq 2\sigma W \sqrt{2 \log(2/\delta) / (t|\mathcal{D}| + \beta B)}$$

Thus, the SAA solution approaches the true optimal face at a rate of $\mathcal{O}(1/\sqrt{t|\mathcal{D}| + \beta B})$, which is faster than the $\mathcal{O}(1/\sqrt{t|\mathcal{D}| + \beta})$ rate when $B = 1$.

**Proof** By Assumption 1 and the union bound, $\|\bar{c}_{t,\beta} - c\|_\infty \leq \sigma \sqrt{2 \log(2p/\delta) / (t|\mathcal{D}| + \beta B)}$ holds with probability $\geq 1 - \delta$. Since $f(\mathbf{v}, \cdot)$ is linear, uniform concentration over the finite polyhedron $\mathcal{V}$ gives:

$$\max_{\mathbf{v} \in \mathcal{V}} |\langle \bar{c}_{t,\beta}, g(\mathbf{v}) \rangle - \langle c, g(\mathbf{v}) \rangle| \leq \sigma W \sqrt{2 \log(2p/\delta) / (t|\mathcal{D}| + \beta B)}$$

The result follows from standard perturbation bounds for linear programs.

## E Optimization Problem and Dataset for Benchmark Experiments

In this section, we will elaborate on the 7 optimization problems in the benchmark experiments, including, for each optimization problem, its real-world production background and significance,

the task of the prediction task part, the specific form of the optimization problem in the decision-making task part, the dataset, and the license.

## E.1 OPTIMIZATION PROBLEM

### E.1.1 BUDGET ALLOCATION

**Problem Background:** Uncertainty-embedded budget allocation problems arise in contexts where nonprofit entities endeavor to propagate philanthropic information across a portfolio of websites, all while operating under the binding constraint of a fixed total budget.

**Prediction Phase:** Given the feature representation $\mathbf{x}^w$ pertaining to website $w$, the task consists of forecasting $\mathbf{y}^w$ namely, the likelihood that informational content hosted on website will successfully reach the intended customer base, as Eq (10).

$$y^w = \mathcal{M}_\theta(\mathbf{x}^w). \tag{10}$$

**Decision Phase:** The overarching objective is to maximize the expected count of users who are reached by at least one website within the set. Mathematically, this optimization challenge is formalized as Eq (11) and Eq (12).

$$\mathbf{v}^*(\mathbf{y}) = \arg\max_{\mathbf{v}} \frac{1}{N} \sum_{u=0}^{N} \left( 1 - \prod_{w=0}^{M} (1 - \mathbf{v}^w \circ \mathbf{y}^{wu}) \right) \tag{11}$$

$$\text{subject to: } \sum_{w=0}^{M} \mathbf{v}^w \leq B \tag{12}$$

**Dataset and License:** The data comes from Yahoo! Webscope Dataset Yahoo (2007) with labels $y_i$ for each user $i$. Access to this dataset is facilitated via Yahoo's publicly accessible data repository , which is expressly designated for non-commercial utilization by academic researchers and scholars. The dataset adheres to Yahoo's stringent data protection protocols, incorporating robust privacy safeguards. Any utilization of the data must strictly conform to the Data Sharing Agreement and the terms of use stipulated by Yahoo .

### E.1.2 CUBIC TOP-K

**Problem Background:** the cubic top-k problem, the task centers on identifying the most salient features embedded within a predictive model, herein referred to as the *top-k* problem.

**Prediction Phase:** Given a dataset $\{\mathbf{x}_i, \mathbf{y}_i\}_{i=1}^n$, where each feature vector $\mathbf{x}_i \sim U[0, 1]$ is drawn from a uniform distribution over the unit interval, the response variable $\mathbf{y}_i$ is generated via a cubic polynomial formulation Eq (13).

$$\mathbf{y}_i = 10\mathbf{x}_i^3 - 6.5\mathbf{x}_i. \tag{13}$$

**Decision Phase:** The objective is to select the $k$-largest elements from the response vector $\mathbf{y}$. Formally, this is defined as Eq (14).

$$\mathbf{v}^\star(\mathbf{y}) = \text{argtopk}_k(\mathbf{y}) \tag{14}$$

where $\text{topk}_k(\cdot)$ denotes the operator that extracts the indices (or values) of the $k$ maximum elements.

**Dataset and License:** Instances are generated via independent sampling from a uniform distribution over $[0, 1)$. For experimental purposes, we configure a dataset comprising $50$ items with a constraint parameter (referred to as "budget") set to $5$. Notably, this dataset is constructed *ab initio* and does not incorporate external data sources from other entities.

### E.1.3 BIPARTITE MATCHING

**Problem Background:** Graph Bipartite Matching, a fundamental problem in social network analytics, findssextensive real-world applications, such as facilitating online job-seeking and friend-recommendation services. A critical challenge emerges when the edge relationships among nodes

are only partially observable or entirely unknown. In such cases, it becomes imperative to perform predictive inference on edge connectivity prior to executing the matching process.

**Prediction Phase:** For each pair of nodes $(i, j)$, given their respective feature representations $\mathbf{x}^i \in \mathbb{R}^d$ and $\mathbf{x}^j \in \mathbb{R}^d$ (where $d$ denotes the dimensionality of the node features), the task is to predict the presence or absence of an edge between these two nodes. We define a binary indicator variable $y^{ij} \in \{0, 1\}$, where $y^{ij} = 1$ signifies the existence of an edge and $y^{ij} = 0$ indicates its absence. The predictive process is shown as Eq (15)

$$y^{ij} = \mathcal{M}_\theta(\mathbf{x}^i, \mathbf{x}^j). \tag{15}$$

**Decision Phase:** The bipartite matching problem can be formulated as an integer linear programming problem subject to permutation constraints. Let $\mathbf{y} \in \mathbb{R}^{N \times N}$ denote the adjacency matrix (which can be either the ground - truth matrix or a predicted one), and $\mathbf{v} \in \mathbb{R}^{N \times N}$ represent the binary matching matrix, where the entry $v^{ij} = 1$ indicates that node $i$ is matched to node $j$. The optimization objective aims to maximize the total matching score while satisfying the permutation constraints, which is formulated as Eq (16).

$$\mathbf{v}^*(\mathbf{y}) = \arg \max_{\mathbf{v} \in \mathbb{V}} \sum_{i=1}^{N} \sum_{j=1}^{N} y^{ij} v^{ij} \tag{16}$$

$$\text{subject to: } \mathbf{v1} = \mathbf{1}, \quad \mathbf{v}^\top \mathbf{1} = \mathbf{1}. \tag{17}$$

In the above, $\mathbb{V} = \left\{ \mathbf{v} \in \{0, 1\}^{N \times N} \mid \text{Eq. (17)} \right\}$ enforces that $\mathbf{v}$ is a permutation matrix. This means that each row and each column of $\mathbf{v}$ contains exactly one non - zero entry (representing a unique matching).

**Dataset and License:** We employ the Cora citation network (Sen et al. (2008)) as the benchmark dataset for graph matching experiments. In this network, each node corresponds to a scholarly paper, and the node features are derived from a bag-of-words representation, resulting in a 1433-dimensional feature vector for each node. We partition the entire graph into 27 sub-graph instances. Each instance consists of 100 nodes and corresponds to a bipartite matching problem with a cardinality constraint of 50 (i.e., there are 50 edges in each valid matching). The Cora dataset is widely used in numerous research endeavors within the domain of graph learning. Based on our verification, it is released under the Creative Commons Attribution 4.0 International (CC BY 4.0) License. This license permits scholarly reuse, modification, and distribution of the dataset, provided that appropriate attribution is given to the original authors.

### E.1.4 SCHEDULING (ENERGY)

**Problem Background:** With the increasing penetration of clean energy sources into the power grid, energy demand profiles and pricing mechanisms have exhibited enhanced adaptability. Within the realm of industrial production, the optimization of scheduling tasks with respect to real - time energy prices holds significant potential for substantial energy conservation and operational cost reduction. This study focuses on formulating and solving an energy-cost-aware scheduling problem, which encompasses both predictive and optimization components.

**Prediction Phase:** The predictive task entails forecasting the energy price for each of the 48 time slots (each slot corresponding to a 30-minute interval) within a given scheduling horizon. To achieve this, a set of relevant features is utilized, including, but not limited to, weather predictions (such as temperature forecasts), wind energy production estimates, and other operational parameters (e.g., production load forecasts). Mathematically, for a time slot index $t$, let $\mathbf{x}_t$ denote the feature vector comprising these relevant attributes, and the goal is to predict the energy price $p_t$, as Eq (18).

$$\hat{p}_t = \mathcal{M}(\mathbf{x}_t). \tag{18}$$

**Decision Phase:** The optimization objective is to minimize the total energy cost associated with scheduling $J$ jobs on $M$ machines, while adhering to the constraints of earliest start time $e_j$ and latest end time $l_j$ for each job $j$.

We define the following parameters: $M$: Number of machines available for job processing. $J$: Number of jobs to be scheduled. $R$: Number of resources required for job execution. $T$: Number of

time slots in a scheduling day (set to $T = 48$ in this study, corresponding to 30 - minute intervals). $e_j$: Earliest start time of job $j$. $l_j$: Latest end time of job $j$. $d_j$: Duration of job $j$. $p_j^t$: Power usage of job $j$ at time slot $t$. $u_{jr}$: Resource usage of job $j$ on resource $r$. $c_{mr}$: Capacity of machine $m$ for resource $r$. Let $v^{jmt}$ be a binary decision variable, where $v^{jmt} = 1$ if job $j$ starts at time slot $t$ on machine $m$, and $v^{jmt} = 0$ otherwise. The objective function aims to minimize the total energy cost of the schedule, which is formulated as a linear program as Eq (19).

$$\min_{\mathbf{v}} \sum_{j \in J} \sum_{m \in M} \sum_{t \in T} v^{jmt} \left( \sum_{t'=t}^{t+d_j-1} p_j^{t'} \right) \qquad (19)$$

subject to the following constraints:

Job Scheduling Uniqueness Constraint: Each job is scheduled on exactly one machine at a unique start time, as shown as Eq (20).

$$\sum_{m \in M} \sum_{t \in T} v^{jmt} = 1, \quad \forall j \in J \notin \mathcal{T}_{jm} \qquad (20)$$

Machine-Job Compatibility Constraint: A job cannot be scheduled on a machine outside the set of available machines for that job, as shown in Eq (21).

$$v^{jmt} = 0, \quad \forall j \in J, \forall m \in M, \forall t \notin \mathcal{T}_{jm} \qquad (21)$$

where $\mathcal{T}_{jm}$ represents the set of valid start times for job $j$ on machine $m$. Time Window Constraint: The job must start after the earliest start time and end before the latest end time, as shown as Eq (22).

$$v^{jmt} = 0, \quad \forall j \in J, \forall m \in M, \forall t + d_j > l_j. \qquad (22)$$

Resource Capacity Constraint: The resource usage of all jobs scheduled on a machine must not exceed the machine's resource capacity, as shown as Eq (23).

$$\sum_{j \in J} \sum_{t'=t-d_j+1}^{t} v^{jmt} u_{jr} \leq c_{mr}, \quad \forall m \in M, \forall r \in R, \forall t \in T. \qquad (23)$$

In our experimental setup, we adopt $N = 3$ machines, $R = 1$ resource, and the resource usage $u_{jr}$ is assumed to be known and constant.

**Dataset and License:** The dataset utilized in this study is sourced from the open-sourced Irish Single Electricity Market Operator (SEMO) dataset (Irinn et al. (2012)). This dataset contains energy-related data collected from midnight 1st November 2011 to 31st December 2013. The energy price prediction task at each time slot is based on a 9-dimensional feature vector, which includes: Calendar attributes (e.g., day of the week, month). Day: ahead weather characteristic estimates (such as wind speed, temperature forecasts). SEMO day: ahead forecasted energy load. Wind: energy production and price forecasts. Actual measurements (including wind speed, temperature, $CO_2$ intensity, and real-time price). The publicly available SEMO dataset (Irinn et al. (2012)) adopted in this research is licensed and regulated by the Commission for Regulation of Utilities (CRU) in Ireland and the Utility Regulator for Northern Ireland (URENI, formerly known as NIAUR). Researchers utilizing this dataset must adhere to the licensing terms and regulatory requirements set forth by these authorities.

### E.1.5 PROFILES

**Problem Background:** Asset allocation stands as a pivotal mechanism in facilitating the circulation of capital across diverse sectors of the economy, thereby playing a crucial role in enhancing overall economic efficiency. In the realm of financial portfolio management, the optimization of asset distribution aims to strike a balance between maximizing returns and mitigating risks, a challenge that has garnered significant attention in both academic research and practical applications.

**Prediction Phase:** The predictive task involves leveraging historical features, such as daily price series and trading volume data, to forecast the daily return $y$ for a set of $N$ stocks. Mathematically,

let $\mathbf{x}_i$ denote the feature vector associated with stock $i$, which may include, but is not limited to, 10-day, weekly, monthly, and annual historical returns, as well as rolling averages over these time windows. The goal is to predict the return $\hat{y}_i$ for stock $i$ on the subsequent trading day, as Eq (24),

$$\hat{y}_i = \mathcal{M}_\theta(\mathbf{x}_i). \tag{24}$$

**Decision Phase:** The optimization objective is to maximize the expected portfolio return while simultaneously minimizing the associated risk. We define the following variables and parameters: $\mathbf{v} \in \mathbb{R}^N$: A vector where $v^i$ represents the fraction of capital invested in stock $i$, with $0 \leq v^i \leq 1$ and $\sum_{i=1}^N v^i = 1$. $\mathbf{y} \in \mathbb{R}^N$: A vector of expected returns for each stock, where $y^i$ is the predicted return for stock $i$. $\lambda = 0.1$: A risk - aversion parameter that quantifies the trade - off between return and risk. $\mathbf{Q} \in \mathbb{R}^{N \times N}$: A positive semi-definite matrix that characterizes the covariance structure between the returns of different stocks. The optimization problem is formulated as Eq (25) and (26).

$$\mathbf{v}^*(\mathbf{y}) = \arg\max_{\mathbf{v}} \mathbf{v}^\top \mathbf{y} - \lambda \mathbf{v}^\top \mathbf{Q} \mathbf{v}. \tag{25}$$

$$\text{subject to: } \sum_{i=0}^N v^i = 1. \tag{26}$$

Here, the first term $\mathbf{v}^\top \mathbf{y}$ represents the expected return of the portfolio, and the second term $\lambda \mathbf{v}^\top \mathbf{Q} \mathbf{v}$ accounts for the portfolio risk, where the covariance matrix $\mathbf{Q}$ captures the interdependencies between stock returns.

**Dataset and License:** The dataset employed in this study is sourced from the publicly available SP500 dataset: Quandl (2022), which contains financial data of 505 of the largest companies in the US market spanning from 2004 to 2017. The feature set for each stock includes historical returns over multiple time horizons (10-day, weekly, monthly, and annual) and rolling averages computed over these periods. In our experimental setup, we set the risk-aversion parameter $\lambda = 0.1$. The publicly accessible dataset utilized in this research is obtained from the specified website. Users must adhere to the terms and conditions outlined in the corresponding data usage agreement for its legitimate utilization.

### E.1.6 KNAPSACK (GEN)

**Problem Background:** The stochastic knapsack problem emerges as a critical optimization framework in scenarios where agents seek to optimize item selection (e.g., cargo loading, resource allocation) under uncertainty. A paradigmatic application arises in sustainable transportation systems, where minimizing energy consumption—an inherently uncertain cost component—drives the need for robust predictive-optimization models. This work formalizes such a knapsack problem with energy-cost uncertainty as a representative case study.

**Prediction Phase:** For a set of $N$ items, the predictive task involves mapping feature vectors $\mathbf{x}^j \in \mathbb{R}^d$ (describing item $j$) to their corresponding values $y^j$. Mathematically, we define a predictive function as Eq (27).

$$\hat{y}^j = \mathcal{M}_\theta(\mathbf{x}^j). \tag{27}$$

**Decision Phase:** The optimization objective is to maximize the total value of selected items while respecting a capacity constraint. Let: $\mathbf{v} \in \{0, 1\}^N$: Binary selection vector ($v^j = 1$ if item $j$ is selected). $\mathbf{y} \in \mathbb{R}^N$: Vector of item values (predicted or ground-truth). $\mathbf{w} \in \mathbb{R}^N$: Vector of item weights. $C \in \mathbb{R}^+$: Total knapsack capacity. The integer linear program is formulated as Eq (28) and Eq (29).

$$\mathbf{v}^*(\mathbf{y}) = \arg\max_{\mathbf{v}} \sum_{j=1}^N y^j v^j \tag{28}$$

$$\text{subject to: } \sum_{j=1}^N w^j v^j \leq C. \tag{29}$$

**Dataset and License:** We generate a synthetic dataset $\mathcal{D}_{\text{synthetic}} = \{(\mathbf{x}_1, y_1), \ldots, (\mathbf{x}_n, y_n)\}$ following established procedures in the literature Geng . Item values are derived from a polynomial

function, as Eq (30).

$$y_i = \left[ \frac{1}{3.5^{\deg}\sqrt{p}} \left( (B\mathbf{x}_i) + 3 \right)^{\deg} + 1 \right] \cdot \epsilon_i \tag{30}$$

where: $\mathbf{x}_i \sim \mathcal{N}(0, I_p)$: $p$-dimensional feature vectors sampled from a standard Gaussian distribution. $B \in \mathbb{R}^{d \times p}$: Parameter matrix encoding true feature-value relationships, with entries $B_{ij} \sim \text{Bernoulli}(0.5)$. $\epsilon_i \sim U(0, 1)$: Multiplicative noise term. $\deg = 4$: Polynomial degree (fixed for experiments).

We fix experimental parameters as: Capacity $C = 30$. Number of items $N = 20$. Item weights $w^j \sim U(3, 8)$.

### E.1.7   KNAPSACK (ENERGY)

**Problem Background, Prediction Phase, Decision Phase** have already been elaborated for the Knapsack (Gen) instance; the primary distinction between the two tasks lies solely in their datasets. **Dataset and License:**   For empirical validation, we adopt the SEMO dataset (Irinn, O'Sullivan, and Simonis 2012), previously utilized in energy-cost optimization research. In this context: Items: Correspond to time slots in the energy market. Values: Energy-cost savings (or profit) associated with each time slot. Weights: Resource usage $u_{jr}$ of each time slot, treated as deterministic inputs.

### E.2   DFL METHODS

In the main text, we have elaborated on the traditional methods under the learning to rank method and our proposed subset sampling method based on sample average approximation; these details will not be repeated here. This section focuses on introducing other DFL methods.

### E.2.1   SPO

In parallel, an alternative research strand has concentrated on adapting subgradient approximation methodologies, originally devised for continuous linear problems, to discrete-valued scenarios. Specifically, the SPO-relax method introduces a relaxation of the original discrete optimization problem and leverages the surrogate SPO+ loss function, first proposed in Mandi et al. (2020). This loss formulation enables the utilization of subgradient-based updates within a backpropagation-compatible paradigm. Mathematically, the SPO-relax loss is defined as Eq (31).

$$\mathcal{L}_{\text{spo}}(\mathbf{y}, \hat{\mathbf{y}}) = -f\big(\mathbf{v}^*(2\hat{\mathbf{y}} - \mathbf{y}), 2\hat{\mathbf{y}} - \mathbf{y}\big) + 2f\big(\mathbf{v}^*(\mathbf{y}), \mathbf{y}\big) - f\big(\mathbf{v}^*(\mathbf{y}), \mathbf{y}\big) \tag{31}$$

### E.2.2   NCE

Mandi et al. (2022) take $\mathbb{S} \setminus \{\mathbf{v}^*(c)\}$ as negative examples and define a noise-contrastive estimation (NCE) loss, as Eq (32).

$$\mathcal{L}_{\text{NCE}}(\hat{c}, c) = \frac{1}{|\mathbb{S}|} \sum_{\mathbf{v} \in \mathbb{S}} \left( f(\mathbf{v}^*(c), \hat{\mathbf{c}}) - f(\mathbf{v}, \hat{\mathbf{c}}) \right) \tag{32}$$

The novelty lies in Eq. (32) being differentiable without solving the optimization problem. Moreover, if solutions in $\mathbb{S}$ are optimal for arbitrary cost vectors, this approach is equivalent to training within a region of the convex hull of $\mathbb{V}$.

### E.2.3   CPLAYER

Agrawal et al. (2019) propose an approach to differentiate through disciplined convex programs (a subset of convex optimization problems used in domain-specific languages). Introducing disciplined parametrized programming (a subset of disciplined convex programming), they show every such program can be represented as composing an affine map from parameters to problem data, a solver, and an affine map from solver solution to original problem solution .

### E.2.4 IDENTITY

Sahoo et al. (2023) propose a hyperparameter-free approach to embed discrete solvers as differentiable layers in deep learning. Prior methods (input perturbations, relaxation, etc.) have drawbacks like extra hyperparameters or compromised performance. Their work leverages the geometry of discrete solution spaces, treats solvers as negative identities in backpropagation, and uses generic regularization to avoid cost collapse. $\mathbf{I}$ is the identity matrix, and the gradient designed in their paper is shown as Eq (33).

$$\frac{\partial \mathbf{v}}{\partial \mathbf{y}} = -\mathbf{I}. \tag{33}$$

### E.2.5 LODL AND DFL

Mandi et al. (2022) propose a novel approach that abandons surrogates entirely, instead learning loss functions tailored to task - specific information. Notably, theirs is the first method to fully replace the optimization component in decision - focused learning with an automatically learned loss. Key advantages include: (a) reliance only on a black - box oracle for solving the optimization problem, ensuring generalizability; (b) convexity by design, enabling straightforward optimization.

### E.2.6 BLACKBOX

When confronted with the dilemma that the map from $\mathbb{C} \to \mathbb{V}$ is either non-differentiable or has vanishing gradients, Pogančić et al. (2019) adopt a remarkably straightforward remedy: they approximate the gradient via linear interpolation. Their surrogate gradient construction is shown as Eq (34):

$$\frac{\partial \mathcal{L}}{\partial \mathbf{y}} = \frac{1}{\lambda} \left[ \mathbf{v} \left( \hat{\mathbf{y}} + \lambda \frac{\partial L}{\partial \mathbf{v}} (\hat{\mathbf{v}}) \right) - \mathbf{v}(\hat{\mathbf{y}}) \right] \tag{34}$$

### E.2.7 2-STAGE

To ensure an equitable comparison, all end-to-end trainable models and the 2-stage baseline share an identical predictive backbone: a compact multi-layer perceptron (MLP). Given an input feature vector $\mathbf{x}$, the predictor $\mathcal{M}$ is defined by the recursive relation:

$$\mathbf{a}^{(1)} = \mathbf{x},$$

$$\mathbf{a}^{(i+1)} = \phi\big(\mathbf{W}^{(i)}\mathbf{a}^{(i)} + \mathbf{b}^{(i)}\big), i = 1, \ldots, K-1,$$

$$\hat{\mathbf{y}} = \mathbf{a}^{(K)},$$

where $\mathbf{W}^{(i)}$ and $\mathbf{b}^{(i)}$ denote the weight matrix and bias vector of the $i$-th layer, respectively, and $\phi(\cdot) = \max(\cdot, 0)$ is the ReLU activation. Throughout the experiments we fix the depth at $K = 3$ and the hidden dimension at 32.

The 2-stage paradigm serves as the standard baseline whenever the coefficients of the downstream optimization task are uncertain and must be forecast. A supervised predictor is trained on the pre-collected dataset $\mathcal{D} = \{(\mathbf{c}_i, \mathbf{y}_i)\}_{i=1}^{N}$ to minimize either the mean square error (MSE) loss as Eq (35).

$$\mathcal{L}_{\text{MSE}}(\hat{\mathbf{y}}, \mathbf{y}) = \frac{1}{N} \sum_{i=1}^{N} \|\mathbf{y}_i - \hat{\mathbf{y}}_i\|^2, \tag{35}$$

or the binary cross-entropy (BCE) loss:

$$\mathcal{L}_{\text{BCE}}(\hat{\mathbf{y}}, \mathbf{y}) = -\frac{1}{N} \sum_{i=1}^{N} \Big[ y_i \log \hat{y}_i + (1 - y_i) \log(1 - \hat{y}_i) \Big].$$

At test time, the inferred coefficients $\hat{\mathbf{c}} = \mathcal{M}_\theta(\mathbf{x})$ are treated as deterministic inputs, after which an off-the-shelf solver is invoked to obtain the final decision.

Notably, the overall training objective in this 2-stage pipeline is entirely dictated by the *prediction* loss (MSE or BCE); no task-specific decision loss is backpropagated.

### E.3 CURATED DATASET

The curated dataset has been made publicly available on the benchmark data open-source framework website by Geng et al. (2024) Since we do not have the permission to repost the cloud storage link of their curated dataset, and this experiment requires the use of Geng et al. (2024) dataset, please refer to their original paper (*Benchmarking PtO and PnO Methods in the Predictive Combinatorial Optimization Regime*) and open-source framework, where detailed instructions can be found in the README file. Additionally, the usage of this dataset is also provided in the README of our experimental program.

## F SUPPLEMENTARY DATA OF THE BENCHMARK EXPERIMENT

**Complete Data.** In the main text, we have presented the some problems. Here, we focus on showcasing the decision and predictive losses and training and testing efficiencies (quantified by the average train and test time. As shown as Table (3-6), Figure (4-7)

**Sensitivity Analysis.** To verify whether the proposed method can maintain its solution performance and stability when prior parameters change, we conduct a sensitivity analysis of the prior parameters. It is important to note that such analysis is problem-dependent and scenario-specific, and practical problems and application scenarios cannot be exhaustively enumerated. Therefore, consistent with the experimental setup in the main text, we select three representative problems of different types and scales from the benchmark for verification and analysis, namely: Knapsack (Gen), Bipartite Matching, and Budget Allocation. Results are shown as Table(7-9) and Figure (8-10) , From the experimental results on problems of different scales, the overall decision quality and training time of the model maintain good stability: although various indicators fluctuate slightly, no severe oscillations occur. This characteristic provides decision-makers with a large degree of freedom in adjusting prior parameters.

| Metrics | 2-stage | Org-Pt | Org-Pr | Org-Lt | SAA-Pt | SAA-Pr | SAA-Lt |
|---|---|---|---|---|---|---|---|
| Avg train Time | 0.086 | 0.933 | 0.360 | 0.365 | 0.365 | 0.365 | 0.475 |
| Avg test Time | 0.529 | 0.600 | 0.567 | 0.556 | 0.611 | 0.572 | 0.594 |
| Dec Loss | 36.667 | 37.333 | 37.833 | 38.500 | 36.167 | 37.651 | 36.500 |
| Pre Loss | 0.654 | 0.588 | 0.229 | 0.634 | 0.634 | 0.592 | 0.661 |

Table 3: Bipartitematching complete data

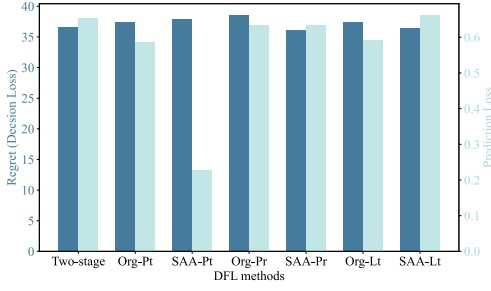
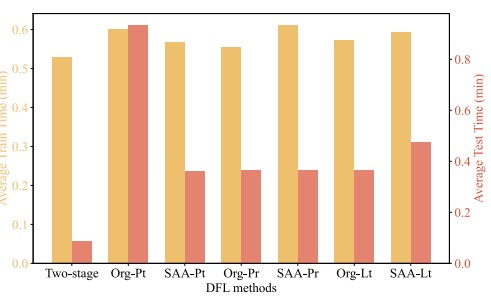

(a) Decision loss and prediction loss      (b) Training and testing time

Figure 4: Taking the Bipartitematching problem as an example, we compare the performance and efficiency with the original model.

| Metrics | 2-stage | Org-Pt | Org-Pr | Org-Lt | SAA-Pt | SAA-Pr | SAA-Lt |
|---|---|---|---|---|---|---|---|
| Avg train Time | 0.002 | 2.719 | 2.157 | 2.891 | 2.581 | 7.983 | 7.448 |
| Avg test Time | 7.056 | 8.916 | 9.143 | 9.881 | 9.312 | 8.999 | 8.758 |
| Dec Loss | 0.076 | 0.031 | 0.073 | 0.009 | 0.008 | 0.007 | 0.005 |
| Pre Loss | 0.001 | 0.006 | 0.017 | 0.035 | 0.032 | 0.004 | 0.003 |

Table 4: Budgetalloc. complete data

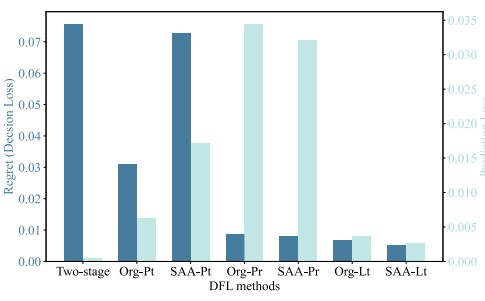
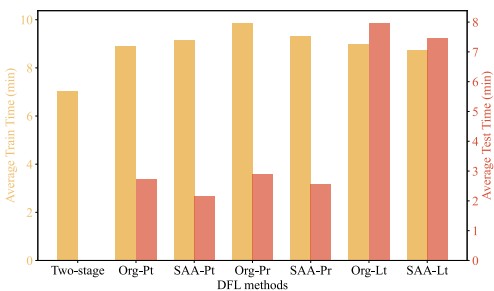

(a) Decision loss and prediction loss

(b) Training and testing time

Figure 5: Taking the Budgetalloc. problem as an example, we compare the performance and efficiency with the original model.

| Metrics | 2-stage | Org-Pt | Org-Pr | Org-Lt | SAA-Pt | SAA-Pr | SAA-Lt |
|---|---|---|---|---|---|---|---|
| Avg train Time | 0.002 | 0.560 | 0.199 | 0.772 | 0.078 | 0.394 | 0.190 |
| Avg test Time | 0.005 | 0.001 | 0.003 | 0.003 | 0.005 | 0.016 | 0.009 |
| Dec Loss | 0.012 | 0.129 | 0.009 | 1.578 | 0.117 | 1.533 | 0.070 |
| Pre Loss | 0.050 | 0.960 | 0.020 | 2.345 | 2.248 | 1.482 | 351.373 |

Table 5: CubicTopk complete data

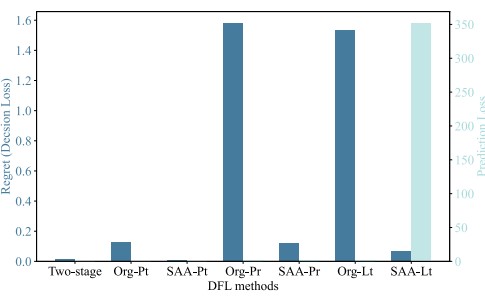
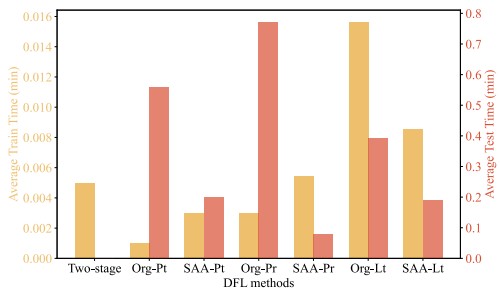

(a) Decision loss and prediction loss

(b) Training and testing time

Figure 6: Taking the CubicTopk problem as an example, we compare the performance and efficiency with the original model.

| Metrics | 2-stage | Org-Pt | Org-Pr | Org-Lt | SAA-Pt | SAA-Pr | SAA-Lt |
|---|---|---|---|---|---|---|---|
| Avg train Time | 0.538 | 0.542 | 0.669 | 0.566 | 0.566 | 0.570 | 0.570 |
| Avg test Time | 0.011 | 0.004 | 0.001 | 0.003 | 0.003 | 0.002 | 0.002 |
| Dec Loss | 72.195 | 91.902 | 80.043 | 84.477 | 92.225 | 66.569 | 65.945 |
| Pre Loss | 648.400 | 791.459 | 851.082 | 4983.956 | 4991.830 | 4724.682 | 4781.894 |

Table 6: Knap. (Ener.) complete data

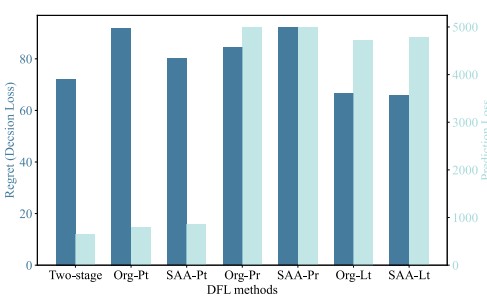

(a) Decision loss and prediction loss

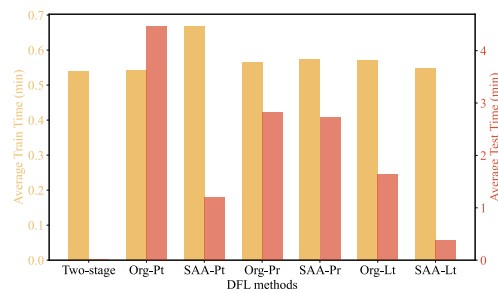

(b) Training and testing time

Figure 7: Taking the Knap. (Ener.) problem as an example, we compare the performance and efficiency with the original model.

| Methods | Metrics | $C = 30$ | $C = 45$ | $C = 60$ | $C = 75$ | $C = 90$ |
|---|---|---|---|---|---|---|
| Baseline | Regret | 2.395 | 2.080 | 1.730 | 1.320 | 1.190 |
| (2-stage) | Avg train T | 0.038 | 0.0440 | 0.003 | 0.046 | 0.045 |
| SAA-Pt | Regret | 2.142 | 2.450 | 2.120 | 2.120 | 2.155 |
| | Avg train T | 0.881 | 2.664 | 2.535 | 2.368 | 1.447 |
| SAA-Pr | Regret | 3.170 | 3.250 | 3.530 | 3.060 | 2.055 |
| | Avg train T | 0.625 | 1.155 | 1.640 | 1.748 | 0.804 |
| SAA-Lt | Regret | 2.370 | 2.025 | 1.950 | 1.535 | 1.245 |
| | Avg train T | 0.771 | 1.397 | 1.396 | 1.843 | 0.470 |

Table 7: Taking the Knapsack (Gen) problem as an example, sensitivity table of decision quality and training efficiency with respect to knapsack. $C$.

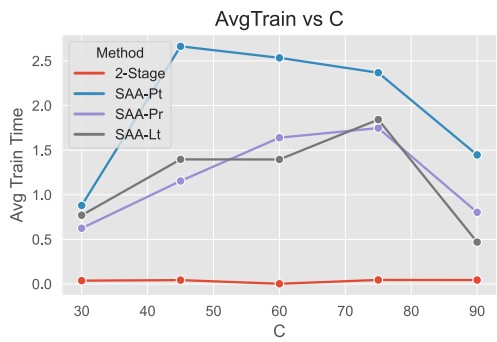

(a) Sensitivity of training efficiency

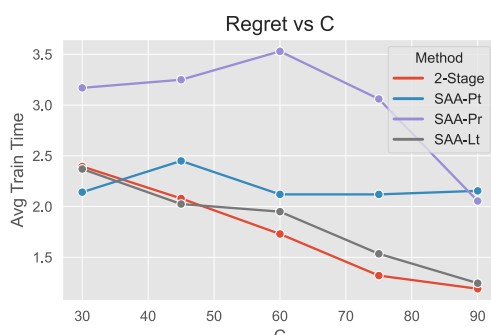

(b) Sensitivity of model performance problem

Figure 8: Model sensitivity on the Knapsack (Gen).

| Methods | Metrics | $N = 5$ | $N = 8$ | $N = 11$ | $N = 14$ | $N = 17$ |
|---|---|---|---|---|---|---|
| Baseline | Regret | 0.076 | 0.071 | 0.067 | 0.083 | 0.073 |
| (2-stage) | Avg train T | 0.002 | 0.002 | 0.003 | 0.003 | 0.003 |
| SAA-Pt | Regret | 0.004 | 0.007 | 0.004 | 0.009 | 0.008 |
| | Avg train T | 2.783 | 2.662 | 2.535 | 2.968 | 3.147 |
| SAA-Pr | Regret | 0.008 | 0.007 | 0.009 | 0.005 | 0.008 |
| | Avg train T | 7.559 | 7.235 | 7.742 | 7.938 | 7.904 |
| SAA-Lt | Regret | 0.005 | 0.007 | 0.006 | 0.010 | 0.007 |
| | Avg train T | 7.243 | 7.509 | 7.623 | 7.995 | 7.622 |

Table 8: Taking the Budgetalloc problem as an example, sensitivity table of decision quality and training efficiency with respect to the number of targets and the number of fake targets $N$.

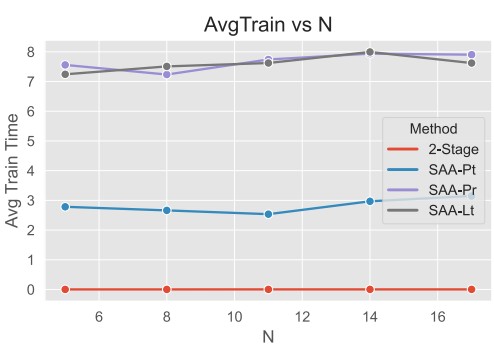 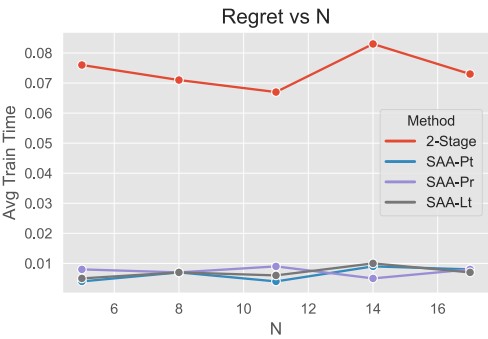

(a) Sensitivity of training efficiency    (b) Sensitivity of model performance

Figure 9: Model sensitivity on the Budgetalloc. problem.

| Methods | Metrics | $N = 25$ | $N = 35$ | $N = 45$ | $N = 55$ | $N = 65$ |
|---|---|---|---|---|---|---|
| Baseline | Regret | 36.932 | 36.978 | 37.757 | 37.557 | 38.557 |
| (2-stage) | Avg train T | 0.088 | 0.081 | 0.077 | 0.082 | 0.089 |
| SAA-Pt | Regret | 36.126 | 36.632 | 36.773 | 37.126 | 37.241 |
| | Avg train T | 0.387 | 0.342 | 0.467 | 0.393 | 0.425 |
| SAA-Pr | Regret | 36.323 | 36.250 | 37.530 | 37.060 | 37.055 |
| | Avg train T | 0.497 | 0.455 | 0.552 | 0.449 | 0.504 |
| SAA-Lt | Regret | 36.034 | 36.025 | 36.950 | 37.535 | 37.245 |
| | Avg train T | 0.372 | 0.402 | 0.399 | 0.453 | 0.430 |

Table 9: Taking the Bipartitematching problem as an example, sensitivity table of decision quality and training efficiency with respect to the number of nodes $N$.

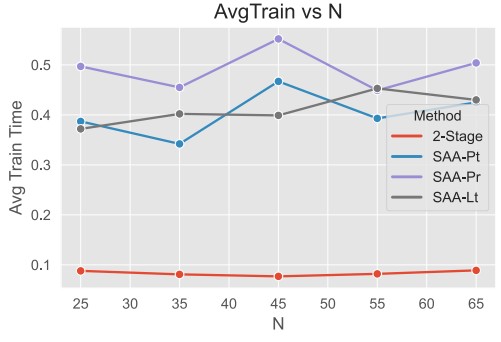 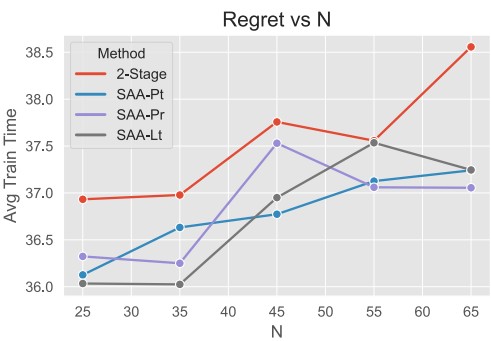

(a) Sensitivity of training efficiency    (b) Sensitivity of model performance

Figure 10: Model sensitivity on the Bipartitematching.

