# OpenReview forum: "Decision-Focused Learning:  Learning to Rank Based on Sample Average Approximation"
_ICLR.cc/2026/Conference — ICLR 2026 Conference Withdrawn Submission_

### Official Review · Reviewer_kLni · 2025-10-26

**Soundness:** 3
**Presentation:** 3
**Contribution:** 2
**Rating:** 2
**Confidence:** 4

**Summary:**

This paper proposes a modification to learning-to-rank approaches for decision focused learning. The main idea is to change the way that solutions are added to the set that the ranking loss is evaluated over, so that instead of adding based on the optimizers of a single-step perturbed cost, instead optimizers of a running-average cost are included.

**Strengths:**

Learning to rank is a promising family of approaches to this problem. The proposed approach is very scalable and easily implementable in existing pipelines.

**Weaknesses:**

My main concern is that the approach is attempting to optimize an internal component of the overall training procedure but it's not clear how well this translates into better results overall. In the results, the relevant comparison seems to be between each Org-* algorithm and the corresponding SAA-*, since this isolates the impact of making the authors' suggested change to the LTR approach. However, it looks like the performance difference from swapping in the new subset construction is very uneven; performance gets worse about as often as it gets better.

Moreoever, claiming that the new method achieves SOTA on 5/7 tasks is based on taking the best across each of the three LTR approaches, which effectively rewards increasing the variance in the performance of the base methods regardless of whether the average/typical performance improves. In order for this to be truly valid, the best of the three LTR losses should be selected on a validation set, not the test set. It would also need to be counted as a computational cost that the "approach" the authors are implicitly proposing is to run all three base losses and select the best one, and the appropriate baselines would then include competitors that take similar ensembling approaches.

Alternately, if the three variants are to be assessed individually, there should be more information about their individual performance levels (e.g., average performance & rank across tasks) since it's not clear whether each of them would outperform their original counterpart in an overall sense (ie the metric right now is implicitly "number of tasks with SOTA result" which is interesting but also leaves out a lot of information compared to "average/typical performance across tasks")

**Questions:**

Comments how we should think about the evaluation (best of three vs evaluating each individually) would be helpful.

---

> ### Author Response · Authors · 2025-11-21
> **Re for the first and third paragraphs in Weakness  and Question**
>
> Thank you very much for your valuable comments (weaknesses and questions). These suggestions have been highly insightful for improving our paper and have helped us achieve progress in multiple aspects. We will respond to each of your concerns in detail.
>
> First, regarding the question you raised, we still adhere to the "best of three" evaluation method. The key reason for this choice is that the design motivation and theoretical foundation of our method are both built on the idea of "selecting the optimal ensemble result"—a key logic that will also serve as the basis for our detailed explanation of Weakness 1 and Weakness 3 you pointed out, as elaborated below:
> ($i$) **Uniqueness of the LTR framework in DFL**: The LTR framework is distinctive among DFL methods, it achieves optimization goals by designing a proxy ranking task while offering multiple optional loss functions. This feature is unique in current mainstream DFL methods and provides a natural premise for the application of ensemble strategies.
> ($ii$)  **Mechanistic differences among multiple loss functions**: As shown in the prediction loss analysis in **Appendix F**, the three loss functions exhibit significant differences in their mechanisms of action for decision optimization. Therefore, selecting the best-performing model from multiple candidates is the optimal ensemble approach tailored to the characteristics of this framework.
> Based on the above premises, we argue that the key goal of the subset construction method should be to **raise the performance upper bound**. As you suggested in the last paragraph of your weakness comment, focusing on the independent performance improvement of individual loss functions can indeed provide more detailed evidence of the method’s effectiveness. However, from the perspective of practical application, this approach is somewhat redundant—only one optimal model can be selected for final deployment, and the performance improvements of other models cannot be truly translated into practical value. Thus, between "the average improvement of the three loss functions" and "an excellent performance upper bound", we believe the latter is more practically meaningful for the LTR framework, with the advantages of lower implementation difficulty and a more direct and concise path.
>
> This design philosophy is precisely our key motivation, which has been elaborated in detail in **Section 3.1 MOTIVATION** and also addresses the question raised in the first paragraph of your weakness comment. Our method is constructed around the goal of "pursuing the ultimate performance upper bound", specifically, the "tightening of the regret upper bound" proven in Theorem 1, which forms the core internal mechanism and theoretical foundation of our approach.
>
> However, this theoretical conclusion cannot guarantee that our method will necessarily outperform or underperform the original method when trained with a single loss function, nor can it ensure improvements in overall average metrics or stability indicators. This is the fundamental reason for the phenomena you observed in the experiments: "the performance difference from swapping in the new subset construction is very uneven; performance gets worse about as often as it gets better", and "effectively rewards increasing the variance in the performance of the base methods regardless of whether the average/typical performance improves".
>
> We acknowledge that this method is indeed overly aggressive and lacks sufficient stability, which constitutes a limitation of our work. To address this, we have added new content in the conclusion section of the paper to explicitly highlight this constraint. Nevertheless, it is important to emphasize that through the "optimal ensemble" strategy, the SAA-LTR framework still achieves highly competitive performance on the benchmark, which is sufficient to realize the effect of "the merits outweighing the flaws".

---

> ### Author Response · Authors · 2025-11-21
> **Re for the second paragraph in Weakness**
>
> Your suggestion in the second paragraph,  the optimal model should be selected on the validation set prior to conducting benchmark tests. It's extremely valuable, as it precisely addresses the gaps in logic and experimental design in the original manuscript. The specific revisions are as follows:Within the DFL method framework, only the LTR framework supports ensemble strategies due to its multiple optional loss functions. Thus, the step of "selecting the optimal model based on the validation set" is exclusively applicable to the LTR framework. We have supplemented the performance of each model under the LTR framework on the validation set in the original manuscript, and in strict accordance with your suggestion, selected the optimal model for subsequent benchmark tests. The detailed results are presented below:
>
> Results of the LTR framework on the validation set:
> | Method| Knapsack(Gen)     | Knapsack (Energy)   |Scheduling   |Budget  Allocation     |TopK (Cubic)        |Bipartite Matching|Portfolio |
> | :--------- | :----------: | :---------: | :---------: | :---------: | :---------: |:---------: |---------: |
> | Org-Pt| 2.894 | 78.123 |19923.463|0.073|0.078|35.500|**0.203**
> | Org-Pr | 2.933      |83.247   |**15823.189**|**0.007**|0.286|35.032|0.233
> | Org-Lt | **2.662** |**77.345** |15572.437|0.008|**0.032**|**34.893**|0.289
> | SAA-Pt| **2.438** | 77.256 | 129955.617|0.064| **0.009**|34.500|0.334|
> | SAA-Pr| 2.773 | 91.002 |38020.166|0.007|0.175|33.250|0.302|
> | SAA-Lt| 2.873 | **76.695** |**22572.967**|**0.006**|0.044|**32.500**|**0.240**|
>
> Results on the benchmark test set:
> | Methods       | Knapsack (Gen) | Knapsack (Energy) | Scheduling (Energy) | Budget Allocation | TopK (Cubic) | Bipartite Matching | Portfolio |
> | ------------- | -------------- | ----------------- | ------------------- | ------------------ | ------------ | ------------------- | --------- |
> | 2-stage DFL   | 6.595          | 8.745             | 1.793               | 20.332             | 0.110        | 92.963              | 0.243     |
> | Blackbox      | 11.744         | 8.353             | 6.272               | 35.970             | 1.974        | 91.364              | 0.380     |
> | Identity      | 24.274         | 35.705            | 5.603               | 26.905             | 13.944       | 91.988              | 0.286     |
> | CPLayer       | 24.769         | 36.402            |**1.505**            | 5.559              | 160.408      | 92.327              | 0.309     |
> | SPO           | 6.223          | 8.407             | 1.786               | 25.700             | 0.172        | 91.006              | 0.245     |
> | NOCL          | 10.448         | 9.567             | 1.663               | 9.979              | 160.412      | 92.113              | **0.160**    |
> | Org-LTR       | 6.031          | 8.083             | 1.540               | 5.742              | 0.193        | 91.035              | 0.367     |
> | SAA-LTR (ours)| **5.907**        | **8.007**             | 2.339               | **4.259**              | **0.082**        |**88.211**             | 0.333     |
>
> For detailed analysis and discussion of the above results, please refer to the revised original manuscript, where all modified and newly added content has been highlighted in red.
>
> Finally, we fully agree with your point raised in the second paragraph of Weakness 2, the computational time cost of the LTR framework should include the training time of the three models and the ensemble time. As elaborated earlier, among mainstream DFL methods, only the LTR framework supports the ensemble strategy with multiple loss functions, and the resulting computational cost disadvantage has existed since the inception of the traditional DFL-LTR framework. Our proposed SAA-LTR method also fails to completely address this inherent issue: we can only guarantee no additional time complexity increase at the theoretical level. At the experimental level, we reduce the size of the ranking subset through the convergence of the SAA solution, thereby improving training efficiency compared to the traditional DFL-LTR and alleviating this disadvantage to a certain extent.  It is important to clarify that the computational cost issue is an inherent limitation of the LTR framework, rather than the core focus of this study. This problem itself is highly challenging: although our method can reduce the average training time, it still needs to complete the training of models corresponding to all LTR loss functions, and thus is inherently constrained by the cumulative training cost inherent in ensemble methods. Therefore, the cumulative training cost brought by the ensemble strategy is an inherent shortcoming of the LTR framework. We have added new content in the conclusion section of the paper to explicitly highlight this limitation.

---

> ### Author Response · Authors · 2025-11-21
> **Re for limitations of our methods**
>
> In response to your comments on the weaknesses, besides the aforementioned revisions, we have added the following statements at the end of the paper to address the limitations of our method:
> **However, our method still has limitations: ($i$) Our LTR-SAA is an aggressive method that pursues the performance upper bound, and it cannot guarantee improvements in average performance or stability across all loss function families; ($ii$) As an ensemble framework based on selecting the optimal model, LTR still suffers from the efficiency disadvantage of requiring additional training and evaluation of multiple models compared to other DFL methods, which our method fails to address.**

---

### Official Review · Reviewer_5Bzz · 2025-10-28

**Soundness:** 2
**Presentation:** 2
**Contribution:** 1
**Rating:** 4
**Confidence:** 4

**Summary:**

The paper presents an incremental improvement to the Decision-Focused Learning (DFL) framework, specifically addressing the ranking subset construction issue in Learning to Rank (LTR) with a Sample Average Approximation (SAA) approach. While the proposed method demonstrates improvements in decision quality and experimental results, it largely remains an extension of existing work, rather than introducing a truly novel concept.

**Strengths:**

- The paper effectively identifies a gap in the existing Decision-Focused Learning (DFL) framework, particularly regarding the subset construction problem within Learning to Rank (LTR).
- The authors provide detailed experimental setups, including datasets, hyperparameters, and code, ensuring that their results are reproducible.

**Weaknesses:**

- While the paper presents a novel way to address the ranking subset construction problem, it can be seen as an incremental improvement rather than a breakthrough in DFL or LTR. The concept of improving ranking subsets has been explored before, and the novelty of using SAA for this purpose does not represent a major conceptual shift in the field.
- The work primarily focuses on **solving a specific subproblem within the broader DFL-LTR framework**. Although this problem is important, the contribution seems more incremental than revolutionary, and it does not fundamentally change the landscape of decision-focused learning methods.
- The paper does not sufficiently explore how the method scales to larger, more complex, or real-world optimization problems. There is no discussion on how the method performs when applied to dynamic systems or problems with uncertainty.
- The paper could benefit from a more detailed sensitivity analysis. A more comprehensive sensitivity analysis would help readers understand the robustness of the method and how it behaves under different conditions.

**Questions:**

- While the theoretical analysis provides performance bounds and regret minimization guarantees, how do these theoretical results generalize to more complex real-world problems? Are there any assumptions in the theoretical model that might limit its applicability to a wider range of problems?
- You claim that the method introduces no additional time complexity and is compatible with existing LTR loss functions. However, could there be any practical limitations when applying this method to larger datasets or problems with more complex decision spaces? How does the method handle outliers or noisy data in real-world settings?
- In the experiments, how does the size of the ranking subset impact performance? Does increasing the size of the subset always improve the results, or is there a point where further increasing the subset size leads to diminishing returns?

---

> ### Author Response · Authors · 2025-11-21
> **Re for Some Issues in Weakness 1**
>
> Dear Reviewer,
> Thank you for your careful review and recognition of our work. We will explain and clarify each of the weaknesses and questions you raised. If there is any overlap between the weaknesses and questions, we will address them together. The order of responses may be slightly interleaved, but all points will be covered, we appreciate your understanding.
>
> Regarding your comment in Weakness 1 that "The concept of improving ranking subsets has been explored before," we believe this statement is inaccurate and would like to elaborate: Since the DFL-LTR framework was proposed by Mandi et al. in 2022, the optimization of ranking subsets has been consistently overlooked. As indicated in our motivation **Section 3.1 MOTIVATION** , the method they employed is essentially a simple solution-set sampling strategy originally designed for DFL methods under the perturbation framework. When directly applied to the LTR framework, it merely enforces ranking subset sampling. Theoretically, there is no basis to explain why two DFL methods with fundamentally distinct theoretical natures can share the same solution-set sampling method, let alone achieve improvement or optimization of performance for the specific framework.  In summary, we argue that within the DFL-LTR framework, subset optimization methods are virtually unstudied, they have never even been systematically discussed or recognized. Our work fills this important overlooked gap. Furthermore, the SAA method has never been applied in the traditional LTR framework, as SAA is inherently a data-driven stochastic optimization method. In traditional LTR, the sets to be ranked are deterministic and finite, with no explicit optimization objectives or directions, thus requiring no further optimization. In contrast, the DFL scenario introduces new optimization needs due to the inability to fully enumerate ranking sets, making the integration of SAA method necessary.

---

> ### Author Response · Authors · 2025-11-21
> **Re for  Weakness 2 and  Some Issues in Weakness 1**
>
> Regarding your comment in both Weakness 1 and Weakness 2 that our method is an "incremental improvement" rather than a revolutionary or breakthrough contribution, we would like to elaborate on the significance of our work from the following two aspects:
> ($i$) Our supplement to the DFL-LTR framework is substantial. Applying LTR loss functions to DFL scenarios introduces a core issue absent in traditional LTR: as shown in Section X, the size of the ranking sets can be infinitely large or grow exponentially with the solution dimension, making full enumeration impossible. Compared to traditional LTR, DFL-LTR requires not only well-designed loss functions but also optimized ranking subsets, these two key components are equally important and indispensable, rather than a "subproblem" as you suggested. Furthermore, while loss function design can still draw on traditional LTR experiences, optimizing ranking subsets has no precedents in traditional LTR and represents an entirely new research task.
> ($ii$) Our work inspires ideas and prospects for integrating DFL methods with data-driven stochastic optimization methods. If raw features are not used as additional decision information, and optimization is performed solely based on labels (i.e., $\min \mathbb{E}[f(\mathbf{v};\mathbf{c})]$), there exist many mature and highly interpretable data-driven stochastic optimization methods, such as the SAA method adopted in this paper and distributionally robust optimization based on Wasserstein distance. Their key advantage lies in enhancing the robustness and statistical interpretability of decisions. However, existing DFL methods have not been integrated with stochastic optimization methods. Ours is the first work to specifically achieve this integration for the LTR framework, yielding promising performance. We believe combining these two data-driven methodologies is a highly promising and underexplored direction. Previously, we only briefly mentioned this direction in the method section; thanks to your insightful comments, we have reconsidered its significance and supplemented relevant discussions in the conclusion section of the original manuscript to inspire future research. The specific supplementary content is as follows: **Nevertheless, it is worth noting that our method also inspires a research direction that combines two data-driven optimization paradigms: DFL and stochastic optimization. This is because DFL can obtain additional raw feature information, while stochastic optimization offers statistical interpretability in decision-making. If the research outcomes of these two currently independent fields can be integrated, it will be an interesting and promising research direction.** All modifications have been highlighted in red in the revised manuscript.

---

> ### Author Response · Authors · 2025-11-21
> **Re for Some Issues in Weakness 3 and Question 2**
>
> For Weakness 3 regarding the "larger scale" concern, and Question 2 asking "However, could there be any practical limitations when applying this method to larger datasets" (i.e., the adaptability to large-scale data), we elaborate on the theoretical and experimental aspects as follows:
> From a theoretical perspective, we infer that your core concern is the potential computational inefficiency caused by larger-scale data. Nevertheless, it is important to clarify that our method does not introduce additional time complexity compared to traditional LTR. Furthermore, as demonstrated in the experimental section, the ranking subsets of LTR-SAA exhibit convergence properties, resulting in a smaller size. For the LTR loss function family where computational complexity is linearly related to the size of ranking subsets, our SAA method achieves superior training efficiency and is more adaptable to large-scale optimization problems than traditional DFL-LTR methods.
> Experimentally, we have validated our method on the latest   open-source DFL benchmark (published at NeurIPS 2024), which includes datasets of small, medium, and large scales. It is worth noting that further verification of the method's performance on even larger-scale data requires dedicated datasets and benchmark tracks. While we could conduct supplementary validation using private datasets, this would reduce the credibility of the results—public benchmark results from conferences at the same level as ICLR are more persuasive.

---

> ### Author Response · Authors · 2025-11-21
> **Re for Some Issues in Weakness 3 and Question 1**
>
> Regarding the "more complex, or real-world optimization problems" mentioned in Weakness 3, and the concerns about the applicability to complex real-world problems raised in Question 1 "how do these theoretical results generalize to more complex real-world problems? Are there any assumptions in the theoretical model that might limit its applicability to a wider range of problems?", we respond from both theoretical and experimental perspectives as follows:  From a theoretical standpoint, we emphasize that within the DFL methodology, the LTR framework has been proven to be one of the most generalizable frameworks. It is applicable to various complex optimization problems, independent of the specific form or type of the problem, whether linear or nonlinear, convex or nonconvex, whether the solution space is discrete, continuous, or a complex combination of both, and even without explicitly defining objective functions and constraints. The generality in addressing complex problems, which you highlighted, is precisely one of the most prominent advantages of the LTR framework over other specific DFL methods. Correspondingly, its excellent generality allows for the use of more complex mathematical forms when modeling real-world optimization problems, thereby capturing and abstracting the key characteristics of real-world problems more accurately, which is exactly one of the key strengths of our method. Relevant details have been elaborated in the introduction and the problem background and description sections.  Experimentally, the benchmark datasets we adopted include real-world data collected from platforms such as Yahoo and Cora, and we have constructed real-world optimization scenarios (e.g., power dispatch and portfolio investment) based on these data, fully verifying the feasibility of our method in practical problems. To further enhance its real-world application value, it also requires the improvement and support of dedicated datasets and benchmark tracks.

---

> ### Author Response · Authors · 2025-11-21
> **Re for Some Issue in Weakness 3**
>
> Regarding your comment in Weakness 3 that "There is no discussion on how the method performs when applied to dynamic systems or problems with uncertainty," and the suggestion in Weakness 4 that "The paper could benefit from a more detailed sensitivity analysis. A more comprehensive sensitivity analysis would help readers understand the robustness of the method and how it behaves under different conditions," we consider these two points to be concerns within the same research direction and respond to them collectively as follows.Measuring a method’s adaptability to dynamic systems and problems with uncertainty, as well as its robustness under different conditions, can be mathematically reduced to analyzing the performance stability of the model when affected by changes in relevant parameters, which is precisely the sensitivity analysis you mentioned. The parameters in the model can be divided into two categories, as elaborated below:
> ($i$) Non-priori parameters: These refer to parameter $\mathbf{c}$ in the objective function $f(\mathbf{v};\mathbf{c})$ of the optimization problem, and they are dynamic and uncertain. It is important to emphasize that the core purpose of DFL methods (or 2-stage methods) is to predict such non-priori parameters through machine learning models to achieve optimized decision-making. Therefore, the development of all DFL theoretical methods and the design of benchmark experiments are centered on improving and verifying the decision performance when non-priori parameters change, and this constitutes the main content of this paper.
> ($ii$) Priori parameters: These parameters are preset before optimization and decision-making, and will not adjust with changes in the decision environment (e.g., the knapsack capacity in the knapsack problem), which are similar in nature to hyperparameters. We acknowledge the significance of conducting sensitivity analysis on such priori parameters and hyperparameters, and have already performed sensitivity analysis on relevant priori parameters for the Knapsack (Gen) problem (a type of optimization problem) in the **Appendix F Sensitivity Analysis**  of the original manuscript. However, it should be noted that such analysis lacks generality, our method does not introduce additional hyperparameters, and the relevant parameters are inherent personalized settings of specific optimization problems. Thus, this targeted analysis can only be presented in the appendix.
> Nevertheless, we agree with your perspective that conducting sensitivity analysis on such parameters facilitates a deeper understanding of the model’s stability. Therefore, in the original **Appendix F**, following the experimental standards of the main text, we have additionally supplemented sensitivity experiments on priori parameters for representative optimization problems (Budget Allocation and Bipartite Matching) under different scales and optimization types. All newly added and modified content has been highlighted in red.  It is important to further clarify that such sensitivity analysis exhibits significant problem dependence and individuality: its design and conclusions are strongly associated with specific optimization problems and lack generality across different problems. Given this characteristic, we have only conducted experiments and analysis on representative problems to strike a balance between effectiveness and conciseness.

---

> ### Author Response · Authors · 2025-11-21
> **Re for Some Issue in Question 2**
>
> Regarding your question in Question 2: "How does the method handle outliers or noisy data in real-world settings," our response is as follows:
> First, it should be clarified that this issue essentially relates to data preprocessing in dataset and benchmark work, which is not the core focus of our research. Assuming the input data to the method has undergone preprocessing, our method actually demonstrates significant advantages in overcoming data outliers and noise compared to the traditional DFL-LTR framework:  As discussed in **Section 3.3 Superiority of rank subset construction via Sample Average Approximation method**, the traditional single-point greedy subset sampling method is susceptible to abnormal features $\mathbf{x}$, leading to the prediction of low-quality and abnormal $\hat{\mathbf{c}}$. This further results in abnormal solutions, and including such abnormal solutions in the subset directly impairs learning performance and reduces efficiency. In contrast, the solutions obtained by our LTR-SAA subset sampling method are all stochastic optimization solutions based on previous samples, which are of high quality and not affected by individual abnormal features.

---

> ### Author Response · Authors · 2025-11-21
> **Re for Question 3**
>
> Regarding the question in Question 3 concerning the impact of ranking subset size on performance and the  diminishing returns, we have provided a qualitative description in **Section 4.2 Training Efficiency** of the experimental part: model performance is independent of ranking subset size. The key reason is that the solutions obtained by our SAA method exhibit global stochastic optimization convergence, resulting in a smaller ranking subset size. Since the computational time complexity of the LTR loss function family is linearly related to subset size, the average training time per epoch is significantly reduced without compromising model performanc, our method still achieves the SOTA in decision quality. The key reason why the SAA method can achieve superior performance with a smaller subset size lies in its ability to approximate the optimal equivalent subset  $\mathbb{S}^\*$  and reduce variance. Model performance is determined by subset quality rather than size, so there is no conclusion that increasing subset size will necessarily improve model performance. Regarding the  diminishing returns by increasing the number of subsets you mentioned, this is precisely the core motivation for us to improve subset quality. As elaborated in the theoretical part: due to the variation of predicted values $\hat{\mathbf{c}}$ in each training round, some solutions in the subset may not belong to the optimal equivalent subset $\mathbb{S}^\*$.  This reduces the accuracy of the loss function in characterizing the effectiveness of the surrogate ranking task, thereby decreasing training efficiency. These solutions are precisely the key cause of the diminishing  returns phenomenon where increasing subset size yields no significant performance improvement. Thus, the diminishing  returns you mentioned essentially corresponds to the boundary point of whether a solution belongs to the optimal equivalent subset  $\mathbb{S}^\*$ in each training round: if a solution is not in $\mathbb{S}^\*$, performance gains will be limited even if the subset size is expanded. However, it is practically challenging to obtain pure $\mathbb{S}^\*$ solutions in each training round. Our SAA method effectively approximates $\mathbb{S}^\*$ by solving stochastic optimization solutions, which alleviates the diminishing  returns of traditional greedy sampling.

---

> > ### Comment · Reviewer_5Bzz · 2025-11-27
> >
> > Thank you for taking the time to provide a detailed rebuttal. I read it carefully, and although some points became clearer, several of my central concerns remain.
> >
> > My first concern is about **innovation**. After reviewing both the paper and the response, I still feel that the method is essentially a *direct combination of existing tools from stochastic optimization and traditional LTR sampling*. The paper does not seem to offer a new modeling perspective or a shift in how we think about DFL-LTR as a whole. The motivation also stays quite close to comparing against classical LTR, which makes the contribution feel narrow and somewhat incremental.
> >
> > My second concern is the **theoretical depth**. The assumptions behind the main theorem are not described with the clarity needed to understand when the result truly applies. The theory also does not extend very far. The complexity discussion remains high-level, and it is still unclear whether the use of SAA introduces extra training or inference costs once the problem size grows beyond the benchmarks.
> >
> > A third point that remains unresolved is the explanation about **noise and outliers**. The rebuttal first treats this as outside the paper’s scope, but then claims the method has advantages in noisy settings. These two statements do not sit well together, and the lack of dedicated experiments makes it difficult to accept the claim of robustness.
> >
> > Finally, the discussion about **subset size and diminishing returns** stays at a qualitative level. The argument relies on intuition rather than on quantitative evidence or a clear definition of what constitutes a “high-quality” subset. Without empirical curves or statistical analysis, the issue feels only partially addressed.
> >
> > I appreciate the additional explanations in the response, and I think they help at the level of presentation. However, the core questions above are still there. With these points in mind, I believe that keeping a score of **4** is reasonable. This does not mean the paper is without value because many stronger papers in this area at ICLR also receive similar or lower scores.

---

### Official Review · Reviewer_hEcy · 2025-10-31

**Soundness:** 3
**Presentation:** 2
**Contribution:** 2
**Rating:** 6
**Confidence:** 3

**Summary:**

This work addresses the limitations of decision-focused learning with rank-based objectives. The authors apply a sample-based averaging mechanism as an approximation to the complete solution set and derive performance upper bounds for the entire framework. Experimental results demonstrate improvements over existing baselines on test open-source benchmarks in terms of both training time and solution quality.

**Strengths:**

1. The work addresses a practical and specific limitation of the DFL framework with rank-based objectives.
2. The paper presents a comprehensive theoretical analysis for the sampling-based learning approach, including PAC regret bounds.

**Weaknesses:**

1. The methodological contribution appears incremental: the work primarily applies well-established stochastic approximation techniques to a specific decision-focused learning problem with rank-based objectives. It would be helpful if the authors could clarify: (a) what novel theoretical insights or algorithmic innovations are introduced beyond standard stochastic approximation, and (b) whether stochastic approximation is already a common approach in learning-to-rank research, and if so, how this work differs from or extends those methods.
2. The overall framework is limited to a specific problem setting, and it is unclear how generalizable this method is to other decision-focused learning scenarios or different types of objectives beyond rank-based ones. A discussion of potential extensions or the fundamental barriers to generalization would strengthen the paper.

**Questions:**

Please refer to the weaknesses.

---

> ### Author Response · Authors · 2025-11-21
> **Re for weakness 1**
>
> Dear Reviewer,
> Thank you for your careful review and recognition of our work. Your comments are profoundly insightful, and we will explain and clarify each weakness you identified as follows:
>
> For Weakness 1 (a):We consider our most innovative contribution to be the PAC-type upper bound for LTR-SAA, as it is the first theory to combine the optimization philosophies of DFL and data-driven stochastic methods, the two research domains that were previously independent.  Our method lies in the first-time establishment and integration of two major data-driven decision optimization paradigms: DFL methods and data-driven stochastic optimization methods within the LTR framework. Prior to this, DFL methods had never considered leveraging statistically optimal stochastic optimization methods such as SAA for decision support, while SAA and similar methods could not utilize raw feature information for decision-making like DFL methods. Our innovative integration of these two mature research fields under the LTR framework not only provides insights for the development of DFL and stochastic optimization methods but also inspires the exploration of new methods (see our response to Weakness 2 for details).
>
> For Weakness 1 (b): The Sample Average Approximation (SAA) method has never been applied in the LTR framework before. This is because SAA is inherently a data-driven stochastic optimization method, and traditional LTR involves deterministic and finite sets to be ranked without explicit optimization objectives or directions, thus requiring no further optimization. In contrast, the DFL scenario introduces new optimization needs due to the inability to fully enumerate ranking sets, leading to the development of our method.

---

> ### Author Response · Authors · 2025-11-21
> **Re for weakness 2**
>
> For Weakness 2: Although our method is currently limited to DFL methods under the LTR framework, it inspires innovative ideas for the integration of DFL methods with data-driven stochastic optimization methods and demonstrates broad prospects for the development of this research direction.  If raw features are not used as additional decision information, and optimization is performed solely based on labels (i.e., $\min \mathbb{E}[f(\mathbf{v};\mathbf{c})]$), there exist many mature and highly interpretable data-driven stochastic optimization methods, such as the SAA method adopted in this paper and distributionally robust optimization based on Wasserstein distance. Their key advantage lies in enhancing the robustness and statistical interpretability of decisions. However, existing DFL methods have not been integrated with stochastic optimization methods. Ours is the first work to specifically achieve this integration for the LTR framework, yielding promising performance. We believe combining these two data-driven methodologies is a highly promising and underexplored direction. Previously, we only briefly mentioned this direction in the method section; thanks to your insightful comments, we have reconsidered its significance and supplemented relevant discussions in the conclusion section of the original manuscript to inspire future research.The specific supplementary content is as follows: **Nevertheless, it is worth noting that our method also inspires a research direction that combines two data-driven optimization paradigms: DFL and stochastic optimization. This is because DFL can obtain additional raw feature information, while stochastic optimization offers statistical interpretability in decision-making. If the research outcomes of these two currently independent fields can be integrated, it will be an interesting and promising research direction.** All modifications have been highlighted in red in the revised manuscript.

---

### Official Review · Reviewer_yacA · 2025-11-01

**Soundness:** 3
**Presentation:** 3
**Contribution:** 3
**Rating:** 6
**Confidence:** 3

**Summary:**

This paper presents a novel framework for Decision-Focused Learning (DFL) using Learning to Rank (LTR) methods and introduces an innovative Sample Average Approximation (SAA)-based ranking subset construction. The primary objective is to enhance decision-making quality in downstream combinatorial optimization problems by refining the selection process of subsets of solutions during training. Specifically, this paper addresses a significant issue in existing DFL-LTR approaches, where the solution space cannot be fully enumerated, leading to inefficiencies in decision quality. By utilizing stochastic optimization over minibatches with the SAA method, the authors propose a plug-and-play subset construction method that does not add extra time complexity or hyperparameters and is compatible with various ranking loss functions.

**Strengths:**

1. This paper addresses a significant gap in decision-focused learning by proposing a dedicated method for ranking subset construction within the LTR framework. This is a crucial contribution for improving model performance in combinatorial optimization problems.

2. The use of the Sample Average Approximation for subset construction is an insightful and effective way to reduce variance and stabilize the training process. The SAA-LTR method avoids the pitfalls of greedy algorithms that rely on potentially noisy or fluctuating predictions.

3. The proposed method consistently outperforms previous DFL methods, achieving state-of-the-art decision quality in 5 out of 7 optimization problems in the benchmark tests, without introducing the cost of additional time complexity.

**Weaknesses:**

1. While Lemma 3 proves the variance of ranking scores decays, the experimental results in Figure 3(a) show that this reduction is incremental and contributes little to performance gains. It is essential to clarify why the primary performance driver appears to be the better approximation of $S^*$ rather than the variance reduction, which was initially considered a key advantage.

2. The use of a cumulative cost $c_{rec}$ over the entire training history is a central but potentially confusing aspect. It is treated as an unbiased estimate of the population cost, but computed from a non-stationary sequence as the model parameters $\theta$ are updated. A more detailed discussion on the justification for this in a non-i.i.d. online learning setting would strengthen the theoretical argument.

**Questions:**

How would the SAA method be adapted for a batch_size > 1?

---

> ### Author Response · Authors · 2025-11-21
> **Re for weakness 1**
>
> Dear Reviewer, Thank you for your careful review and recognition of our work. Your comments are incisive and insightful. We will address each of the weaknesses and questions you raised point by point, and clarify the additional supplements and revisions made to the original manuscript as follows:
>
> Regarding Weakness 1:
> In response to your question about why approximating $\mathbb{S}^\*$ in experiments is more beneficial for performance improvement than variance reduction, we provide a comprehensive analysis combining theoretical and experimental evidence as follows:
>
> Theorem 1 explicitly decomposes the upper bound of regret into two components:
> $$
> regret(T) \leq \underbrace{\frac{2\sigma G W }{\sqrt{|\mathcal{D}|}}\sqrt{\frac{2 \log(2/\delta)}{T}}}\_{\text{subset error}} + \underbrace{\mathcal{O}\left( \frac{G^2 \log |\mathbb{V}|}{\sqrt{T}} \right)}\_{\text{learning error}}
> $$
>
> As observed from the decomposition:
> The key function of variance reduction is to stabilize gradients, but it does not directly reduce either term in this decomposition. Thus, it cannot directly tighten the regret upper bound at the theoretical level. Our experimental results further confirm this: the magnitude of variance reduction is limited, leading to an extremely marginal improvement in overall performance.
>
> In contrast, approximating the optimal subset $\mathbb{S}^\*$ can directly reduce the "subset error" term in the decomposition, and our experimental results have verified that its approximation effect is significant.
>
> In summary, combining theoretical derivation and experimental validation, approximating $\mathbb{S}^\*$ indeed contributes more to tightening the regret upper bound, thereby being more conducive to performance improvement.
>
> Thank you for your valuable suggestion! We agree that we should callback Theorem 1 from the theoretical section in the experimental analysis and provide in-depth interpretation combined with the theory. To this end, we have supplemented the corresponding analysis of the above experimental results and theoretical explanations in **Section 4.2 Verification of ranking score variance reduction and approximation to the $\mathbb{S}^{\*}$**. All supplementary content has been highlighted in red in the revised version.

---

> ### Author Response · Authors · 2025-11-21
> **Re for weakness 2**
>
> Regarding Weakness 2:
> Thank you for raising this subtle yet core question. Below, we elaborate on why $\mathbf{c}_{\text{rec}}$ remains unbiased and theoretically valid in the online, non-stationary environment:
>
> ($i$) **Essential definition of $\mathbf{c}\_{\text{rec}}$**: $\mathbf{c}\_{\text{rec}}=\sum\_{t'=1}^{t}\sum\_{i=1}^{\beta} \mathbf{c}\_i$, where $\mathbf{c}\_i$ denotes the true historical labels (fixed once the data is sampled) rather than predictions based on the current learnable parameter $\theta\_t$. The update of $\mathbf{c}\_{\text{rec}}$ does not participate in the data generation process nor feed back into the training loop. Thus, the construction of $\mathbf{c}\_{\text{rec}}$ always adheres to the independent and identically distributed (i.i.d.) premise in Assumption 1, and is irrelevant to $\theta\_t$ at any moment during training.
>
> ($ii$) **Unbiasedness of $c\_{\text{rec}}$ for the population under dynamic updates of $\theta\_t$**: Although the learnable parameter $\theta\_t$ evolves non-stationarily with the iteration step $t$, $\mathbf{c}\_i$ satisfies the i.i.d. assumption. Regardless of the update path of $\theta\_t$, $\mathbf{c}\_{\text{rec}}$ remains an unbiased estimate of the population cost at each step $t$.
>
> Thank you for your valuable question! To facilitate readers' understanding, we have added the following note after Assumption 1 (the relevant content has been highlighted in red in the revised manuscript):
> **It is important to note that based on the assumption in Assumption 1 that historical labels $\mathbf{c}\_i$ are independently and identically distributed (i.i.d.), the following conclusion can be derived: although the learnable parameter $\theta\_t$ evolves non-stationarily, the recommended cost estimate $\mathbf{c}\_{\text{rec}}$ remains an unbiased estimate of the population cost at each epoch $t$.**

---

> ### Author Response · Authors · 2025-11-21
> **Re for question**
>
> Regarding your question:
> Your question is of great practical value. Although traditional DFL-LTR defaults to **batch size = 1** (sample-by-sample update), we initially adopted this setting to align with the experimental protocol of the baseline method. However, our proposed LTR-SAA method itself has no dependence on the batch size. Below, we illustrate its applicability in the scenario of **batch size > 1** from both implementation and theoretical perspectives:
> $(i)$ **Seamless integration and straightforward implementation**: Assuming the batch size is $B$, the update operation is adjusted to: $c\_{rec} += c\_{batch}$ (where $c\_{batch}=\sum\_{i=1}^B c\_i$); the rest of the solution process remains identical to that when batch size = 1.
> $(ii)$ **Inheritability of theoretical advantages**:
> - For **Lemma 3 (Variance reduction via cumulative minibatch)**:
>   $\operatorname{Var}[\langle\bar{\mathbf{c}}\_{t,\beta},g(\mathbf{v})\rangle] = g(\mathbf{v})^\top \Sigma g(\mathbf{v})/(t|\mathcal{D}|+\beta )\leq \sigma^2 \|g(\mathbf{v})\|^2/(t|\mathcal{D}|+\beta )\\xrightarrow{\text{B>1}}\\operatorname{Var}[\langle\bar{\mathbf{c}}\_{t,\beta},g(\mathbf{v})\rangle] = g(\mathbf{v})^\top \Sigma g(\mathbf{v})/(t|\mathcal{D}|+\beta B)\leq \sigma^2 \|g(\mathbf{v})\|^2/(t|\mathcal{D}|+\beta B) .$
>   It can be seen that the decay rate of the rank loss variance increases from $\mathcal{O}(1/(t|\mathcal{D}|+\beta ))$ (when $B=1$) to $\mathcal{O}(1/(t|\mathcal{D}|+\beta B ))$ (when $B>1$), meaning the variance decreases faster.
> - For **Lemma 4 (Solution error bound)**:
>   $\|g(\mathbf{v}^{\text{SAA}}\_{t,\beta}) - g(\mathbf{v}^\text{true})\| \leq 2\sigma W \sqrt{(2 \log(2 / \delta))/(t|\mathcal{D}|+\beta)}\\xrightarrow{\text{B>1}}\\\|g(\mathbf{v}^{\text{SAA}}\_{t,\beta}) - g(\mathbf{v}^\text{true})\| \leq 2\sigma W \sqrt{(2 \log(2 / \delta))/(t|\mathcal{D}|+\beta B)}$
>   Thus, the SAA solution converges to the true optimal face at a rate improved from $1/\sqrt{t|\mathcal{D}|+\beta }$ (when $B=1$) to $1/\sqrt{t|\mathcal{D}|+\beta B}$ (when $B>1$), i.e., the convergence to the true optimal face is faster.
>
> Thank you again for this highly practical question! We have provided a detailed supplement to this issue in **Appendix D.3** of the revised manuscript, and the relevant content has been highlighted in red.

---

> > ### Comment · Reviewer_yacA · 2025-11-24
> >
> > Thanks for your responses. Some concerns are addressed (e.g., weakness 1). However, due to formatting issues, I failed to understand the answers to the other questions.

---

> > > ### Author Response · Authors · 2025-11-24
> > > **We have revised the format of Re and offer our sincere apologies for the inconvenience caused**
> > >
> > > We sincerely apologize to the reviewers for overlooking the format differences between local Markdown and OpenReview Markdown, which caused display errors. We have revised all our original responses, and kindly request you to review our revised replies again. We deeply apologize for any inconvenience this may have caused.

---

### Note · Authors · 2026-02-01

I have read and agree with the venue's withdrawal policy on behalf of myself and my co-authors.

---

### Meta-Review · Area_Chair_MRT8 · 2026-01-04

**Summary:**

The paper proposes an SAA-based ranking subset construction approach for decision-focused learning under a learning-to-rank framework, supported by theoretical analysis and experiments on several benchmark tasks. However, reviewers raise several key concerns.

- Reviewers question the novelty and conceptual contribution, viewing the method as largely an incremental combination of established stochastic approximation ideas with existing DFL-LTR formulations, without a clear new modeling perspective or a decisive algorithmic insight beyond prior approaches.

- Multiple reviewers expressed concerns about the generalization ability and parameter sensitivity of the proposed framework.

- Reviewer kLni specifically questioned the reasonableness and fairness of the experimental settings used in the paper.

**Reviewer Concerns:**

After the rebuttal, some concerns are clarified (e.g., additional explanations and limited supplementary results). However, the main concerns remain unresolved. In particular, the work is perceived as incremental, the evaluation protocol does not cleanly isolate the contribution from model selection across multiple LTR losses, and the empirical evidence is not yet strong enough to substantiate the claimed mechanism and robustness/generalization under more realistic conditions.

**Reviewer Scores:**

The reviews show meaningful disagreement. Two reviewers are generally positive, while the other two express negative assessments. However, even the more positive reviewers indicate that several of their concerns remain unresolved. In particular, the negative reviewer 5Bzz explicitly states that the core issues persist. As a result, I expect that most reviewers would maintain their original scores.

---

### Decision · Program_Chairs · 2026-01-26

Reject